# Polyaniline/Biopolymer Composite Systems for Humidity Sensor Applications: A Review

**DOI:** 10.3390/polym13162722

**Published:** 2021-08-14

**Authors:** Yuriy A. Anisimov, Richard W. Evitts, Duncan E. Cree, Lee D. Wilson

**Affiliations:** 1Department of Chemistry, University of Saskatchewan, 110 Science Place (Room 156 Thorvaldson Building), Saskatoon, SK S7N 5C9, Canada; yuriy.anisimov@usask.ca; 2Department of Chemical and Biological Engineering, University of Saskatchewan, 57 Campus Drive, Saskatoon, SK S7N 5A9, Canada; richard.evitts@usask.ca; 3Department of Mechanical Engineering, University of Saskatchewan, 57 Campus Drive, Saskatoon, SK S7N 5A9, Canada

**Keywords:** humidity sensor, polyaniline, biopolymer, hydration, adsorption, electrical conductivity, response/recovery time, hysteresis, sensing mechanism, hydrophilicity/hydrophobicity

## Abstract

The development of polyaniline (PANI)/biomaterial composites as humidity sensor materials represents an emerging area of advanced materials with promising applications. The increasing attention to biopolymer materials as desiccants for humidity sensor components can be explained by their sustainability and propensity to absorb water. This review represents a literature survey, covering the last decade, which is focused on the interrelationship between the core properties and moisture responsiveness of multicomponent polymer/biomaterial composites. This contribution provides an overview of humidity-sensing materials and the corresponding sensors that emphasize the resistive (impedance) type of PANI devices. The key physicochemical properties that affect moisture sensitivity include the following: swelling, water vapor adsorption capacity, porosity, electrical conductivity, and enthalpies of adsorption and vaporization. Some key features of humidity-sensing materials involve the response time, recovery time, and hysteresis error. This work presents a discussion on various types of humidity-responsive composite materials that contain PANI and biopolymers, such as cellulose, chitosan and structurally related systems, along with a brief overview of carbonaceous and ceramic materials. The effect of additive components, such as polyvinyl alcohol (PVA), for film fabrication and their adsorption properties are also discussed. The mechanisms of hydration and proton transfer, as well as the relationship with conductivity is discussed. The literature survey on hydration reveals that the textural properties (surface area and pore structure) of a material, along with the hydrophile–lipophile balance (HLB) play a crucial role. The role of HLB is important in PANI/biopolymer materials for understanding hydration phenomena and hydrophobic effects. Fundamental aspects of hydration studies that are relevant to humidity sensor materials are reviewed. The experimental design of humidity sensor materials is described, and their relevant physicochemical characterization methods are covered, along with some perspectives on future directions in research on PANI-based humidity sensors.

## 1. Introduction

To advance the field of multicomponent polyaniline (PANI)/biomaterial composites for humidity-sensing devices, a review of past studies on related developments enables the identification of knowledge gaps in the field. This contribution provides an overview of recent studies that highlight humidity-sensing PANI materials and their key structural and physicochemical properties. The mechanism of polyaniline-water vapor interaction will be presented along with physicochemical properties of composites related to pore structure, flexibility, hydrophilicity, surface area, conductivity and others. Suitable synthetic methodologies will enable the design of new (bio)polymer systems from known, accessible and naturally abundant components. This information can assist in advancing practical moisture-sensitive systems via a bottom-up approach, by incorporating tunable properties to yield improved materials for use in moisture-measurement systems. This review will address several objectives, as follows:To provide a literature overview of PANI/biopolymer sensor materials;To provide insight on the moisture-sensing mechanism of PANI and its hybrid composites that contain biopolymer systems;To develop a rational approach for the design of PANI-based materials that are suitable for humidity-sensing applications.

Figure 1a shows a statistical survey of the literature related to PANI-based humidity sensors over the last decade. The total number of research works (about 100) is limited, which suggests that this field is positioned well for further study and future development. Figure 1b shows a statistical survey of various materials that could serve as potential humidity sensors. In contrast to conventional ceramics [1,2,3,4,5,6,7,8,9], biopolymer materials are highly promising since they are (a) sustainable and environmentally friendly, (b) low cost, and (c) relatively easy to prepare. The search results, shown in Figure 1a, used the following keyword search terms: “polyaniline + humidity sensors”, whereas Figure 1b employed “polyaniline + biopolymer + composites”.

Some authors refer to PANI/CHT composites to as “chitaline” or “PANI-*g*-CHT” and will be used interchangeably herein. The relevance of “chitaline” materials relates to the uncertainty in the structure of such PANI/CHT composites, which will be discussed later. Figure 1 illustrates that chitosan and cellulose are among the most commonly investigated biopolymers used in conjunction with PANI. Currently, there is no humidity sensor device based on PANI/CHT technology. However, there is one known humidity sensor based on chitin, the native biopolymer precursor of chitosan [10], and some sensors are based on PANI/CLL composites [11,12,13]. For other PANI/biopolymer composites, there is a dearth of information, and limited information on such humidity sensors is available in the open scientific literature. Hence, there is a need to provide coverage of such materials in line with the objectives stated above.

The regulation of humidity plays a significant role in the maintenance of human health and providing comfortable conditions in many aspects of daily life. Relative humidity (RH) is related to the ratio of the partial pressure of water vapor to the total pressure of the saturated vapor at a given temperature and is often expressed in %. Humidity control within indoor environments, such as commercial and residential buildings, plays a role in maintaining the quality of infrastructure and consumable goods. For example, moisture can affect the stability of paintings, computers, plants, food, fruits, vegetables, egg incubation, coatings, chemicals, etc. Moisture-responsive materials have been tested and designed over the decades to monitor humidity in order to prevent the degradation of various items. They range from a simple human hair to sophisticated ceramics [1,2,3,4,5,6,7,8,9] or conductive polymer materials [10,14,15,16]. Among these systems, advances in the development of conductive organic polymers have resulted in lightweight, compact, and cost-effective materials that have widened the conventional scope of application beyond traditional ceramic and carbonaceous materials [17]. In this category, PANI has promoted ongoing research interest due to its intrinsic conductivity, where it can be used exclusively as a humidity-responsive material. However, to enhance its properties, PANI can be prepared in the form of composites by the addition of biopolymer additives, such as starch [18,19,20], cellulose [21,22,23] and chitosan [24,25,26].

A humidity sensor is synonymous with a hygrometer, where early hygrometers were based on human hair due to its ability to stretch and shrink according to the changes in the humidity [27]. About half a century ago there were popular “psychrometric charts”, which looked like two-dimensional tables with a “dry” and “wet” thermometer reading [28]. Modern humidity sensors are electronic devices that measure both moisture content (RH, %) and temperature, as these two physical quantities are linked. There is no need to compare thermometer readings as the value of RH (%) readily appears on an electronic display, due to the alteration of electrical currents. The aim of such technology is to precisely detect changes in humidity and convert them into RH values. Humidity sensors can be divided into three types, as outlined in Table 1.

As shown in Table 1, the most appropriate sensors for general daily life are resistive and capacitive types since they operate at room temperature and allow the measurement of RH with good accuracy. Resistive sensors are practical, cost-effective, and suitable for everyday use, and can be subdivided into two categories: polymer-based sensors and ceramic sensors [30]. Capacitive sensors possess the ability to measure RH with higher precision, but they are relatively expensive. Thermal sensors are applicable for industrial use in high-temperature conditions. While ceramic sensors are popular and widely used, polymer-based humidity sensors are used less frequently, in spite of the lower manufacturing cost of polymer-based sensors. In turn, this has driven the interest in further research and development for these materials.

The moisture-sensing mechanism is generally based on the diffusion and mobility of hydrogen (H^+^) or hydroxide (OH^−^) ions (also known as the Grotthuss mechanism) [31]. Water is an amphoteric molecule that can dissociate into H^+^ and OH^−^ ions, especially onto the surface of moisture-responsive materials. Hydrogen ions are attracted by the electron-rich O-atoms, which can further interact with water molecules and transform them into hydronium (H_3_O^+^) ions:H_2_O + H^+^ → H_3_O^+^(1)

H^+^ will then be released into the liquid layer located on the adsorption surface layer, and the cycle will repeat continuously:H_3_O^+^ → H_2_O + H^+^(2)

In this case, hydrogen ions are the dominant carriers of the electric charge on the superficial aqueous layers [32]. Another possible mechanism of the electric charge transport process is also based on the transmission of hydroxide ions [33]. When a water molecule eliminates H^+^, it contributes to the protonation of an adjacent H_2_O molecule, resulting in the formation of OH^−^, as follows:2H_2_O → H_3_O^+^ + OH^−^(3)

Then, the OH^−^ ion interacts with the next water molecule, inducing the chain of the OH^−^ transmission process:OH^−^ + H_2_O → H_2_O + OH^−^(4)

The mechanism described above demonstrates that electrical conductivity is a key requirement for moisture-responsive materials. In order to attract water molecules, the material should be hydrophilic and can be prepared in the form of a thin film as part of the design of a humidity sensor device [34].

The moisture-responsive nature of a material used in a resistive (impedance) humidity sensor can be divided into three categories, based on the most abundant material component: polymers, ceramics and polymer/ceramic materials, as outlined in Table 2.

Table 2 represents three classes of material types (polymers, ceramics and polymer/ceramic) [1,2,3,4,5,6,7,8,9,38,39,40,41,42,43,44,45,46,47,48,49]. Additional classification becomes more complicated as some materials can be attributed to multiple classes of materials. For instance, classification by a conduction mechanism (electronic or ionic) is not that obvious since most materials exhibit both mechanisms, which depend on their formulation or ambient RH level [50]. A typical humidity sensor may include a polymer (conducting or electrolyte) and inorganic compound (salt, oxide, metal, perovskite etc.), or their combinations may function via an electronic–ionic sensing mechanism. Detailed consideration of Table 2 reveals that polymer components may be classified as biomaterials or synthetics, which can be used in conjunction with a conductive polymer (e.g., PANI/CHN [10]) or a ceramic component (e.g., cellulose nanofibers/ZnO [42]). The use of biopolymers has attracted greater interest due to their relative availability, sustainability, and low cost. Recently, humidity sensors based on chitin [10], cellulose [51,52], chitosan [44,53], starch [54,55,56], cyclodextrin [57] and carrageenan [58] have been reported.

Table 2 highlights some examples of PANI-based humidity sensors reported in the literature, where such sensors are found to consist of variable components: (1) PANI/biopolymer; (2) PANI/metal; (3) PANI/metal oxide; other various permutations and combinations of components are possible. Recent review articles [59,60] enumerate a few dozen types of humidity and other PANI-based sensors. Such types of PANI-based devices are characterized by various parameters: (1) response time; (2) recovery time; and (3) RH interval, in which they have functional properties. The optimal response and recovery times are those having characteristics below 1-min intervals across variable RH (0 to 100%). Sensor materials are not always functional across the full RH range, where many fall within a narrow band or the 5–95% range. For example, a PANI/Co(0) humidity sensor performed with an 8 s response time and 60 s recovery time, while the electric conductivity was dependent on the magnitude of water vapor adsorption [61].

Table 2 provides limited coverage of ceramic humidity sensors since they are among the most abundant and widespread sensor types reported, as described in some recent reviews [31,62,63]. While ceramics are not the focus of the current study, the sensing mechanism that is drawn from the field of ceramic materials has relevance to polymer-based humidity sensors.

The development of a suitable humidity sensor requires a specific material composition to achieve humidity sensitivity behavior and high electrical conductivity. A possible synthetic polymer candidate is PANI, whose electric conductivity has been demonstrated widely in other studies [59]. In conjunction with a suitable hydrophilic polymer, PANI can be tailored as a candidate material for humidity sensors. Biopolymer components, such as cellulose, starch, or chitosan, may serve a structural role or act as the hydrophilic counterpart in such sensors. An important and necessary condition is that the material should be prepared as a thin film that has suitable film-forming characteristics, which is a key feature of humidity-sensing device fabrication.

Polyaniline is an interesting synthetic polymer with unique properties that have been known for more than 150 years; its facile synthesis occurs via an oxidation process [64]. Multiple factors are known to influence the electrical conductivity of PANI, which affords tailoring of its properties for sensing applications. PANI has numerous oxidation (colored forms; cf. Figure 2 and Table 3) and protonated forms (pK_a_ values) that endow variable electrical conductivity for diverse sensor applications.

The half-oxidized form of PANI (emeraldine) is the most stable and is chiefly used for PANI-based sensor materials, where grey shading is used to emphasize its importance. Both leucoemeraldine and pernigraniline forms are unstable and exist only under certain conditions. Among the forms of PANI, only the PANI-ES is electrically conductive [65], and this constitutes the main structural form for PANI-based humidity sensors herein, unless noted otherwise.

The electrical conductivity depends on the surface area and porosity [62], and various dopant anions, such as chloride, sulfate, polystyrene sulfonate, etc. [68]. In addition, water vapor affects conductivity [69], and that enhances the efficiency of such PANI-based sensor materials.

The synthesis of PANI in variable morphological forms, such as nanosheets [70], nanofibers [71], nanorods [72], nanocoils [73], nanotubes and nanoflowers [74], have been reported, where the morphology depends on the nature of the synthetic conditions. Some structural forms possess high crystallinity with high conductivity, which lends to their utility for electronic devices, including humidity sensors. For example, a higher melting temperature leads to greater chain disorder, whereas ordered polymers are typically obtained at lower temperatures or at slower rates of reagent addition. The reaction time is a key parameter for the preparation of PANI materials. For example, the role of stirring for 48 h and longer has led to the formation of highly agglomerated units, such as nanosheets and nanoflowers [75].

As outlined above, PANI can form hybrid composites with various biopolymers that contain oxygen and nitrogen heteroatoms, such as cellulose, chitosan, starch, and alginate, which can enhance the hydrophilicity and conductivity of the material. Conductivity depends on RH, where the molar fraction of the biopolymer plays a key role in the electrochemical performance of the composite material. This creates a synergistic effect that is due to the π-π * conjugation of PANI, which can affect sensitivity behavior [76]. The synergism between PANI and a second component (mainly the metal ions) has been reported by Singh and Shukla [59]. Surface interactions are critical for producing a sufficient electrical signal for humidity sensing. Judicious selection of biopolymer components and their ratios may yield high-quality humidity sensors with minimal response and recovery times. In some cases, a third matrix component can be incorporated, as outlined for a ternary humidity sensor material that contains PANI/cellulose/PVA [12].

To devise a good humidity sensor, several conditions should be considered. Firstly, the composition of the material may include a conductive polymer (PANI) and a hydrophilic (bio)polymer (such as CHT). Secondly, the materials should display sensitivity to humidity, and the components should be characterized prior to the fabrication of the sensor. Thirdly, the components should have moderately high conductivity, which suggests the utility of materials with lower resistivity values. Such criteria as moisture response and conductivity can be investigated for the pristine components and compared with their blends or composites. While there are no commercial PANI/biopolymer sensors available on the market at the time of writing this review, there are many types of commercial ceramic-based sensors available (cf. Table 2 for selected examples). To further the discussion of PANI/biopolymer composites, an outline of the important design features for good humidity sensors is required. Once the key design elements are outlined, they can be translated to PANI/biopolymer systems to achieve *state-of-the-art* humidity sensor materials, in line with the objectives of this review.

## 2. Key Features of Ceramic Humidity Sensors

In Table 2, ceramic humidity sensors are mainly constituted by metal oxides, perovskites, sulfides, or combinations of such systems. The effective detection of vapor phases is based on their fundamental physicochemical properties. Firstly, the ability of vapor to adsorb onto a ceramic surface is feasible, based on the surface chemistry of the adsorption sites, where the magnitude of adsorption can vary over a range of humidity values. Secondly, the structure of a humidity-sensing material is unique, due to its surface area and pore structure, according to grain size and grain boundaries. Thirdly, the electrical (e.g., conductivity/resistivity, capacitance) and mechanical (i.e., Young’s modulus) properties are subject to change under variable degrees of water vapor adsorption, especially the electrical properties [63].

### 2.1. Water Vapor Adsorption vs. Electrical Properties

The value of water vapor adsorption is proportional to the environmental RH [77]. Hereafter, an increase in RH implicitly assumes an increase in the extent of adsorption, where there are two types of humidity-sensing (resistive and capacitive) devices, as outlined in Table 1. The conductive mechanism for resistive sensors involves electronic or ionic mechanisms or a combination of these effects (cf. Table 2). This implies that the electric charge is born by electrons and/or ions that result in greater conductivity (decreased resistivity) of a sample with greater RH, according to the Grotthuss mechanism (cf. Equations (1)–(4), Section 1). For capacitive sensors, the magnitude of capacitance grows with increasing RH. Their mechanism is different from the resistive one and is based on changes in the dielectric constant of a material upon altering RH [78]. Typical resistance/impedance, vs. RH or capacitance, vs. RH curves, are shown in Figure 3.

Figure 3c shows an exponential decrease of resistance and growth of capacitance that is RH-dependent. Notably, the capacitance of a resistive-type sensor stays constant over the wide range of humidity. In addition, the use of higher alternating current frequencies led to lower resistance and capacitance.

### 2.2. Pore Structure, Grains and Grain Boundaries

A pore is a cavity located inside the grain (intragranular pore) or a space between the different grains (intergranular pore). A grain is a single structural unit of material, whereas the grain boundary is a surface that separates grains from the intergranular space. Pores, grains and grain boundaries form the unique structure of a material and determine its electrical and mechanical properties [82]. A network of intra- and intergranular pores represents a system with interpenetrating capillary channels that can potentially accommodate absorbed water molecules, together with the electrical charge carriers (electrons and ions). Therefore, the presence of pores is vital for effective humidity sensor materials, whereas the pore sizes should be in a range that does not compromise the material mechanical stability. Figure 4 below represents two sensing materials of similar composition, with variable grain and pore structure [83].

Figure 4 highlights an illustrative example of how the porosity of a sample can be tuned. Pristine magnesium chromite has smaller grains (hence, longer grain boundaries) and smaller intergranular pores (Figure 4A); whereas the same sample, modified by TiO_2,_ acquires larger grains, shorter grain boundaries and larger intergranular pores. Intragranular pores are not visible in this image; however, they were measured by mercury intrusion porosimetry. This system exhibits a trend that is similar to intergranular pores [83], where the resistivity of composites was reported to depend on the TiO_2_ content. In turn, the addition of TiO_2_ boosts the resistivity ca. 200-fold, whereas the further addition of TiO_2_ results in a resistivity drop by ca. three orders of magnitude (cf. Table 4).

Conductivity is calculated as an inverse resistivity, which shows a reverse trend in values. The grain boundary is proportional to the surface area, which becomes shorter when the grains and pores increase in size. These experimental data lend support that larger pores constitute a capillary network that allows for the transfer of larger amounts of water, and a larger number of charge carriers, which results in greater conductivity overall.

In summary, the presence of larger pores yields higher conductivity. The concept of tunable “pore size–conductivity” can be translated from ceramics to PANI/biopolymer humidity sensor materials, along with the creation of particle grain interfaces through the presence of defects in the structure of the composite material.

## 3. Structure and Physicochemical Properties of PANI/Biopolymer Composites

### 3.1. PANI/Chitosan Binary Composites

Chitin is a naturally abundant biopolymer derived from the exoskeletons of crustaceans, insects, and fungal biomass. The controlled deacetylation of CHN using chemical or enzymatic processes yields CHT. CHT is a biocompatible, non-toxic, eco-friendly, biodegradable and relatively low-cost biopolymer. Due to its limited mechanical strength in the dry state, CHT materials can be prepared as composites, along with other polymers such as PANI. CHT contains numerous functional groups (hydroxyl and amine) that can readily undergo further covalent modification or alteration via composite formation with PANI.

PANI/CHT binary composites are normally prepared by the chemical polymerization of aniline, with subsequent grafting onto chitosan in acidic media [84,85,86,87,88]. Since the conductivity of PANI depends strongly on its oxidation state, synthesis is often directed at the preparation of conductive emeraldine salt (PANI-ES) forms. Salt formation is aided by the use of dopants such as HCl or acetic acid, where some studies report the use of an intermediary neutralization step with NaOH (leading to a de-doped emeraldine base (PANI-EB) form), with subsequent re-doping upon the addition of an acid to obtain conductive PANI-ES [77,85,86]. Polymerization is carried out in the presence of an ammonium persulfate (APS) initiator. In general, aniline and powdered CHT are combined and treated with APS; however, one study reports the use of a prefabricated solid-phase CHT film that was submerged in an aniline-APS solution [89]. A generalized synthetic procedure is illustrated in Figure 5.

Although numerous papers reported comparable synthetic procedures, the structural features of PANI/CHT composites are variable. The bonding interactions between PANI and CHT have been described in the literature [77,90,91,92,93,94,95,96,97], which include noncovalent (hydrogen bonding) interactions shown in Figure 6a, along with covalent grafting of chitosan units onto PANI via covalent bonding (cf. Figure 6b). Such grafted composites (PANI-g-CHT or chitaline) have been reported for some years, where the first record dates back to 1989 [94], and were later reintroduced in 2009 [96], 2013 [97] and 2016 [91]. Notably, the authors based their interpretation on FTIR spectral results. In 2009 [96], ^1^H NMR results were reported, but there was no correlation of the signatures with covalent bonding via the NH group (Figure 6b).

Therefore, the predominant structural form of PANI/CHT (cf. Figure 6a,b) remains an open question, where future research will establish the structure of such composites according to the nature of the synthetic conditions.

It is critically important to know about the structural features of such PANI/CHT composites since hydration processes are inferred to impact on the nature of hydrogen-bonded versus covalently bound systems. During the humidity-sensing process, a water molecule that emerges amidst the H-bonded network will tend to form its own hydrogen bond interactions with PANI or CHT. In turn, this will result in an alteration of the PANI/CHT hydrogen-bonded structure. The dissociation of the hydrogen bonds of PANI/CHT will create variable accessibility for water on the (bio)polymer surface sites and pores upon the influx of water, which may result in interpenetrating capillary channels. These interpolymer channel structures will facilitate electron and proton transfer within the composite, leading to greater conductivity. In contrast, networks containing covalent bonds are static relative to H-bonded systems, which are less amenable to the influx of water. This implies that covalently grafted composites may possess fewer capillary channels and reduced conductivity. Therefore, the hydrogen-bonded network structure in Figure 6a is likely a more favorable ensemble to achieve the desired properties for a humidity sensor material.

The morphology of composites may vary depending on the type of dopant. For PANI/CHT copolymers, several doping acids have been used (acetic acid, CH_3_COOH; hydrochloric acid, HCl; and sulfuric acid, H_2_SO_4_) [84,85]. The morphology of such composites was investigated using scanning electron microscopy (SEM), as shown in Figure 7. Figure 7 reveals that the morphology is dependent on the type of Brønsted acid used during synthesis. In the case of the H_2_SO_4_ dopant, the PANI/CHT composite surface appears to be coarse and random [84], while the surfaces using HCl and CH_3_COOH dopants allowed for better-organized structures. Samples prepared in HCl acquired forms with a granular morphology and a high surface area [85], whereas the use of CH_3_COOH as the dopant resulted in a rose-leaf form of PANI/CHT composite [84]. A complex or rough surface morphology by the judicious choice of acid during synthesis leads to enhanced textural and adsorption properties of samples, as a result of the variable textural properties due to acid-driven templation effects.

The electrical conductivity of PANI/CHT composites is influenced by their morphology, crystallinity, and water vapor adsorption properties. The morphology affects the electrical conductivity of the composites. For example, PANI/CHT composites were prepared in powder and film forms, where various authors used SEM to describe how the morphology was related to the conductivity [84,85,87,89]. These studies revealed that the size and distribution of particles for a powdered sample influenced conductivity, where the smoothness and presence of pores and cracks in films played an important role. A smaller particle size leads to greater conductivity [84], whereas surface irregularities cause a decrease in conductivity [85]. In general, the literature indicates that PANI/CHT films consist of highly smooth surfaces with few defects, which supports their utility for the fabrication of smart devices and sensors. The structure of PANI/CHT composites possesses a semi-crystalline structure, where the degree of crystallinity of PANI/CHT composites is known to affect their electrical conductivity. Crystallinity determines the spatial arrangement of ions in the lattice, which also affects the ion-pair formation and the mobility of charge carriers [98]. Increased crystallinity leads to samples with higher conductivity [84]. The crystallinity of various PANI/CHT composites ranges from 3.4 to 11%, which differs according to the mass ratio of components, dopant counter-ions and substituted/unsubstituted PANI.

XRD results revealed that all PANI/CHT composites generally show a characteristic, sharp XRD line at 2θ-values between 20° and 26°, which relate to the amorphous regions of CHT and PANI, respectively [84,85,89]. The degree of crystallinity is normally calculated as the peak area ratio (area of the crystalline domains/(crystalline area + amorphous area) [84]. Previous reports [85,89] for PANI/CHT composites indicate that such materials possess higher crystallinity with respect to pristine PANI but are lower than that of pure CHT. The electrical conductivity values of PANI/CHT samples have been reported in the range of 10^−6^ to 10^−3^ S/cm [77,84,87]. The water vapor adsorption by PANI/CHT composites shows an increase as the RH levels rise, resulting in greater electrical conductivity. Since PANI/CHT is a binary composite, the addition of the component that adsorbs water results in greater adsorption of water. By comparison, if the additive component does not absorb water, the extent of water uptake may not rise incrementally [77]. Although there are limited reports on PANI/CHT humidity sensors, there was a study outlining a sensor based on PANI and chitin [10]. CHN differs from CHT, due to the presence of N-acetyl groups at C-2, as compared with variable levels of deacetylation at C-2 (glucosamine), according to Figure 8. In other words, the degree of acetylation for chitin is 100%, while that of chitosan is generally 50% or lower. Therefore, research on PANI/CHN composites is also relevant to PANI/CHT materials as a result of their structural similarity, along with PANI/cellulose composites, due to the variable functionality at C-2 for these biopolymers (cf. Figure 8).

The thermal stability (decomposition temperature) of PANI/CHT composites falls in a range-between PANI and CHT. For example, decomposition of PANI chains and rupture of benzene rings occur above 400 °C [87,100], while the degradation of chitosan chains takes place at ca. 290 °C [101]. The decomposition of PANI/CHT copolymers occurs at 330–350 °C [77,87]. The increased stability of copolymers can be attributed to the formation of intra- and intermolecular hydrogen bonds between PANI and the functional (amino, hydroxyl, acetyl) groups of chitosan [102]. On the other hand, stability also depends on the relative content of each (bio)polymer in the composite. For example, greater PANI and less CHT yield a PANI/CHT composite that will decompose at a temperature closer to that for PANI (400 °C). Similarly, greater CHT content in the composite and less PANI results in a decomposition temperature nearer to that of CHT (290 °C). This trend concurs with theoretical expectations, based on a linear combination of the thermal properties of each polymer, according to their respective heat capacity.

### 3.2. Humidity Sensing Mechanism of PANI/CHT Composites

The electrical resistance of PANI/CHN composites decreases linearly with a rise in humidity [10]. This suggests that the electrical conductivity exhibits a linear growth since the conductance is inversely proportional to resistance. A rise in electrical conductivity results from the greater mobility of the dopant ion, which is related to the unfolding polymer chains of PANI. In turn, the curling and uncurling (association/dissociation; cf. Figure 6a) of polymer chains depend on the level of swelling. For instance, when the polymer is hydrated (at a high RH), it physically swells, where the coiled polymer chains uncoil to allow for the transport of an electric charge. CHT and chitin are known to swell tremendously [103,104,105], which enhances the unfolding of polymer chains in the composite. This may suggest that, to a certain degree, the addition of CHT or CHN to PANI will increase its conductivity for wet conditions, despite the non-conductive nature of CHT. Excess amounts of CHT will attenuate the overall conductivity of the composite. The geometry of polymer chains becomes more favorable for the charge transfer, as the (bio)polymer chains are aligned in parallel, which leads to greater conductivity [106]. The linear response shape (conductivity versus % RH) is advantageous, as it can be used in amplifiers (circuits that increase the amplitude of the electric signal) for converting measurands into the measurable values of RH.

On the other hand, elevated humidity delivers water molecules to PANI chains and results in the greater intrinsic conductivity of PANI. This is achieved by the interchain electron transfer, as well as by the greater mobility of dopant ions [107]. As mentioned above, the polymerization of aniline occurs with the aid of a strong oxidant, ammonium persulfate (APS), and therefore PANI contains oxidized, half-oxidized and non-oxidized (reduced) units. Oxidized units are characterized by the quinoid forms (=N–; Q), whereas the reduced fraction consists of benzenoid units (–NH–; B), and the half-oxidized PANI is a mixture of Q/B forms (=N– and –NH–) [108]. Nitrogen atoms serve as the source of available electron pairs, which accounts for the p-type doping of PANI. Lone electron pairs on N-groups of PANI become protonated, according to Equations (5) and (6):–NH– + H^+^ → –NH_2_^+^–(5)
=N– + H^+^ → =NH^+^–(6)

The electron transfer model [109,110,111,112,113] shows that the electron is recurrently hopping between the oxidized and reduced forms of PANI (although it seems to be proton hopping; the proton cannot transfer without interaction with donor electrons). Supporting evidence is provided by NMR results [113], which indicate that such electron transfer occurs in the presence of water:–NH_2_^+^– + H_2_O → –NH– + H_3_O^+^(7)

Mohamed et al. reported that the addition of CHT to PANI changed the ratio of B:Q units (in favor of Q units) [100]. The Q-units contribute to a more linear and rigid structure of PANI/CHT composites that may enhance the proton-hopping mechanism laterally between the chains of PANI and CHT (cf. Figure 9).

Figure 9 illustrates two possible ways of proton transfer within the composite—along the polymer chain (“hop along”) and between the polymer chains (“hop across”). It can be realized more fully if the polymer chains of both PANI and CHT are oriented in parallel, which implies a higher fraction of Q forms in the PANI backbone.

The hydrogen bond network between PANI and CHT accounts for the important role of absorbed water in the humidity-sensing mechanism for PANI. PANI/CHT composite materials are prone to absorbing larger amounts of water versus pristine PANI [77], despite their ability to serve as suitable humidity sensors. Overall, the sensing mechanism can be described as the electron transfer that occurs by the proton exchange, via H_2_O.

Currently, there are no known PANI/CHT humidity sensors available commercially. The impetus for the fabrication of such sensor materials relates to their low cost, structural versatility, and sustainability. In 2011, Li et al. fabricated an H_2_ gas humidity sensor [93]. The PANI/CHT nanocomposite sensor was exposed to various levels of hydrogen in an air stream, where they observed a change in resistance. The authors concluded that a hydrogen-sensing mechanism was supported by H_2_O-polymer interactions, which were caused by the reaction between H_2_ and O_2_ from the air. The response and recovery times of such sensors were slow. However, this feature is commonplace for PANI/biopolymer sensors, as discussed in Section 4.1 below. In general, data from [93] relate to the abovementioned sensing mechanism (cf. Figure 9), which suggests that PANI/CHT composites warrant further investigation.

### 3.3. PANI/Cellulose Binary Composites

Polyaniline-cellulose composites are attractive materials for electrical device applications, due to their high electrical conductivity and amenability for the fabrication of conductive nanocomposites. They can be produced from different types of cellulose, such as microcrystalline cellulose [114,115], nanocellulose [116,117], bacterial cellulose [118,119], cotton cellulose [120,121] and various cellulose derivatives (e.g., cellulose acetate [122,123], carboxymethyl cellulose [13,124] and others [125]). Similar to chitosan, cellulose can form composites that are stabilized by hydrogen bonding and other electrostatic interactions [126]. This leads to the functionalization of PANI to afford new forms of polymer composites. As in chitosan, PANI/cellulose composites are prepared via the oxidative polymerization of aniline using APS in an acidic media [127].

#### 3.3.1. PANI/Bacterial Cellulose Composites

Bacterial or microbial cellulose has gained attention for humidity sensor use over the last decade [118,119,128,129,130,131,132,133,134,135,136]. This type of cellulose is derived from the bacteria cultivated under appropriate conditions. Bacterial cellulose (BC) is attractive due to its outstanding mechanical properties, porosity, and electrical conductivity [137]. For instance, H. Kim et al. [128] produced PANI/BC composites with a conductivity that reached 0.02 S/cm, which is almost 10 to 10^4^ times greater than the same value of PANI/CHT composites [77,87,92]. In another study, Alonso et al. [129] also compared the electrical conductivity of PANI/BC membranes synthesized via in situ and ex situ polymerization. The authors concluded that in situ polymerization favored composites with higher PANI contents, leading to materials with higher conductivity. SEM results for pure BC and PANI/BC composites are shown in Figure 10 [129].

Figure 10a displays bacterial cellulose in its pristine state after drying. It can clearly be seen that pure BC contains an intricate network of nanofibers connected by physical joints. The gaps between individual fibers are much larger than their fiber diameter, which illustrates their outstanding porosity. After modification by PANI, the fiber network surface is altered, as shown in Figure 10b. The newly formed composite has a unique morphology resembling that of native BC, but mostly consisting of granules and flakes. The surface of a composite becomes more compact, and pores are being obstructed by incorporated PANI, which implies a decreased distance between the PANI and BC units. The SEM results support the hypothesis that favorable interactions occur between the functional groups of BC and PANI, in line with the role of hydrogen-bonding effects (cf. Figure 10 and Figure 11a).

Similar to chitosan, cellulose-containing samples exhibit hydrogen bonding between BC and PANI (cf. Figure 11a). However, this was concluded based on Fourier transform infrared (FTIR) results, which are likely to yield unequivocal interpretations on the structure of the composite fiber structure [114,119,126]. Some complementary methods (UV-Vis, PXRD and TGA) were also applied, but serve to provide indirect evidence of chemical interactions, in contrast to the use of NMR spectroscopy, as outlined in one published report indicating that the “… composite has the characteristic features of both polyaniline and cellulose” [126]. Another group performed the grafting of cellulose onto PANI using epoxychloropropane as a cross-linker. They propose the structure shown in Figure 11b, according to spectral characterization by FTIR and X-ray photoelectron spectroscopy (XPS) [118].

The thermal stability of PANI/BC composites correlates with the IR spectral data, where the spectra of PANI/BC composites show weak interactions between –OH groups of cellulose and –NH groups of PANI. In addition, weak interactions between the β-glycosidic oxygen of BC and –NH groups of PANI also provide support for a hydrogen bond formation that is similar to the results for PANI/chitosan composites [92]. Interestingly, hydrogen bond formation between PANI and BC causes weakening of the intermolecular hydrogen bonding between cellulose chains, as reflected in the thermogravimetry profiles for PANI/BC [138]. The onset degradation temperature of the binary composites resulted in lower values than that of the pristine cellulose (224 °C vs. 276 °C). Nevertheless, the thermal stability of PANI/BC composites is still sufficiently high to favor humidity measurements in ambient conditions. The relatively high electric conductivity of PANI/BC indicates the potential suitability of such composites for humidity sensing.

#### 3.3.2. PANI/Carboxymethylcellulose (CMC) Composite

A PANI/CMC composite material was developed as a working humidity sensor [13]. The key parameters for humidity sensor performance were given: the response time (10 s) and recovery time (90 s) were reported between RH 25–75%. At RH levels above 75%, the response became saturated. The properties for PANI/CMC compare favorably with PANI/chitin humidity sensor materials [10]. The response time was ca. 30 s, with a measurable humidity that ranged between 10% and 100% RH, where the humidity absorption mechanism for the PANI/CMC composite occurred via the curling and uncurling of the (bio)polymer chains, [13] similar to that described for chitosan above (cf. Section 3.2).

The structure of a generalized PANI/CMC composite is sketched in Figure 12a, where it can be seen that it is similar to PANI/BC and PANI/CHT composites in terms of the hydrogen bonding arrangements between carboxyl or carboxylate groups of CMC and the NHR groups of PANI [124]. These interactions were confirmed by IR spectroscopy, similar to structurally related biopolymers, such as cellulose and chitosan. The authors have pointed out that CMC (sodium salt) requires greater (alkaline) pH to prompt its greater dissociation into –COO^−^ and Na^+^. After that, the negatively charged carboxylate anion is electrostatically attracted to the positively charged anilinium cation during the polymerization process, resulting in a greater conductivity of the PANI/CMC composites. Indeed, the conductivity of pristine PANI (6.3·10^−4^ S/cm) proved to be twice as low as a PANI/CMC composite (12.5·10^−4^ S/cm) [124].

### 3.4. PANI/CLL Ternary Composites

A remarkable humidity sensor, consisting of PANI, nanofibrillated cellulose (NFC), and PVA, was reported in 2019 [12]. This composite has limited properties in the case of PANI, such as hygroscopicity and flexibility, which improved upon the addition of NFC and PVA, respectively. Similar to other PANI/CLL composites, it forms a hydrogen-bonded structure that likely becomes more complex upon the addition of PVA. Based on IR spectral data, Anju et al. [12] proposed that the PANI chains are sandwiched between NFC and PVA chains, according to the following illustration (cf. Figure 13).

The humidity-sensitive mechanism occurs due to the intercalation of water molecules between the chains of PANI and PVA. The hydrogen bonds of PANI-PVA break, with the subsequent formation of PANI-H_2_O and H_2_O-PVA hydrogen bonds. The important features of this sensor are discussed further (cf. Section 4.1).

### 3.5. PANI/Starch Binary Composites

PANI/starch (PANI/STR) composites are normally synthesized via the in situ polymerization of aniline with starch under acidic conditions [139,140,141]. The synthesis is similar to that for PANI/CHT and PANI/CLL composites, meaning that various PANI/biopolymer composites can be obtained by the same method reported for PANI/CHT systems. However, other studies report a different method using post-polymerization of PANI, blended with starch solutions [142,143,144]. The latter is an ex situ method that is in contrast to the conventional approach via the in situ polymerization of PANI with biopolymers.

The structure of PANI/STR composites is portrayed in Figure 12b. It can be seen that the amylose and amylopectin units of starch are connected to PANI via hydrogen bonding, similar to chitosan and cellulose. Such types of interactions were confirmed by an FTIR spectral study, where a detailed spectral assignment for the respective PANI/STR signatures was reported [139]. Interestingly, this study reported an FTIR characterization of PANI/CHT and PANI/CMC composites, since starch, cellulose and chitosan possess certain structural similarities, viz. OH-groups, pyranose rings and glycoside linkages. The reason for comparing PANI/biopolymer signatures with those of pristine PANI in its leucoemeraldine (LE) form (according to a reported ratio of aniline:oxidizer = 1:1.25 [139]) is unclear, whereas the preparation of PANI was reported in its emeraldine (EB) form [145].

SEM images revealed that PANI/STR composites exhibit greater porosity versus the PANI/CMC and PANI/CHT composites [139]. This may lead to higher water vapor adsorption, with an improved sensing performance of the starch-based composites. However, the humidity-sensing properties of PANI/STR materials [139,140,141,142,143,144] have not been widely reported in the open literature.

### 3.6. Concluding Remarks

PANI/biopolymer composites share certain similarities, both in the modality of their preparation and in the corresponding physicochemical properties. The literature mainly covers the in situ polymerization for preparation of PANI/biopolymer composites, whereas ex situ methods were reported for composites that contain cellulose and starch. It appears that both methods lead to hydrogen-bonded materials, while there are some limited reports that indicate the role of covalent grafting between PANI and the biopolymer system. It remains unclear whether there is a particular way to direct a certain type of composite structure, but the occurrence of an H-bonded network is important for humidity-sensing properties. This relates to the ability of water molecules to undergo dynamic formation and the dissociation of hydrogen bonds between PANI with the biopolymer system as a function of variable humidity.

There are some reports of working PANI/polymer humidity sensors described in the literature. This includes a range of PANI/biopolymer systems, as follows: PANI/CHN humidity sensor (2010) [10]; PANI/CMC humidity sensor (2016) [13]; and a recent 2019 report [12] for a comparative study of humidity sensing by ternary composites (PANI/nano- fibrillated cellulose/PVA and PANI/carbon nanofibers/PVA). The latter will be discussed in further detail in Section 4. Overall, the research and applications of humidity sensors based on PANI/biopolymer composites are still at the early stages of research. However, it is noteworthy to compare such PANI/biopolymer systems with other types of well-known conventional humidity sensors, such as carbon-based materials. For example, PANI/carbon humidity sensors represent a variation on the conceptual design of PANI/biopolymer systems to form biopolymer-based sensors, as outlined in Table 5.

As seen from Table 5, these are two extreme cases of moisture-responsive materials, which serve the same goal—to enable humidity sensing. Section 4 will provide an outline of carbon-based humidity sensors.

## 4. Carbon-Based Humidity Sensors

Carbon nanomaterials are represented by the various types of carbon allotropes, such as nanotubes and nanofibers, as well as graphene and carbon black. These materials stand out due to the superior properties that make them perfect candidates for various electronic devices, including humidity sensors. Carbon nanomaterials can acquire various structural hierarchies: one-dimensional (fibers), two-dimensional (films) and three-dimensional (monoliths) shapes that can be used for humidity sensing. Among their various properties that support their utility in the design of sensor materials are their large specific surface area, which allows for the design of sensor materials with high sensitivity and quick response times [146].

In a similar manner, ceramic-, carbon-, and biopolymer-based sensors can work according to resistive or capacitive transduction principles [147,148]. Similar to ceramic sensors, carbon-based capacitive sensors dominate over the resistive-type sensors in terms of flexibility and response/recovery times [146]. Carbon materials can be modified in a similar way to PANI by the addition of hygroscopic and conductive polymers, to yield hybrid materials for humidity sensing. The top features of the most relevant examples for PANI/carbon and PANI/biopolymer sensors are discussed in the section below (cf. Section 4.1).

### 4.1. PANI/Carbon vs. PANI/Biopolymer Sensors

Multiwall carbon nanotubes (MWCNT) have been used in combination with PANI to fabricate and characterize a humidity sensor [149]. Another example of a sensor reported for the detection of humidity and ammonia was fabricated using carbon nanofibers (CNFs), PANI and PVA [12]. Other types of gas sensors were recently reported that include an H_2_S sensor based on carbon aerogel (CA) and PANI [150], and a CO_2_ sensor comprising graphene (G)/PANI [151].

Table 6 illustrates several key features of carbon- and biopolymer-based sensors:Response time (how rapidly the sensor responds to humidity);Recovery time (how rapidly the sensor returns to its initial state after response);Hysteresis (how the sensor recovers fully after each measurement).

In Table 5, biopolymer-based sensors exhibit competitive properties that compare favorably to carbon-based sensors, despite the differing properties of these materials. Nanofibrillated cellulose sensors show a slightly slower response when compared with CNFs, whereas some biopolymer sensors show an even faster response (such as chitin- and CMC-based materials). One conspicuous drawback of biopolymer-based sensors is that they have greater hysteresis error, as evidenced by their incomplete recovery after a humidity measurement.

Figure 14a–c illustrates the hysteresis loops of carbon- and biopolymer-based composites. A hysteresis loop is shaped by the adsorption profile (lower line) and the desorption profile (upper line) evident in Figure 14a,b. In Figure 14c, a wider hysteresis loop is noted that reveals the sample recovers to a lesser extent, and there is a greater hysteresis error. Figure 14 shows a gradual increase in the level of hysteresis from (a) to (b) to (c) at 1%, 5%, and 18%, respectively. The level of hysteresis (H_e_, %) was calculated by either Equation (8) or (9):(8)$$H_{e}=\pm \frac{\Delta R}{2\Delta R_{0}}*100\%\;$$
(9)$$H_{e}=\pm \frac{\Delta C}{2\Delta C_{0}}*100\%\;$$

The difference in resistance (Δ*R*) or difference in capacitance (Δ*C*) depends on the resistive/capacitive sensor type between adsorption and desorption values. The full-scale resistance (Δ*R*_0_) or capacitance (Δ*C*_0_) range of a measurement is given in [149].

It is noteworthy that after multiple work cycles, the hysteresis error increases in Figure 14c, which indicates sensor fatigue [10]. Hence, durability is another important feature of a desirable sensor.

Nanocarbon materials generally possess higher porosity than conventional biopolymers. For example, carbon aerogel materials possess enormous porosity, whereby such types of sensors exhibit an extremely fast response and very slow recovery (cf. Table 6) [150].

As discussed in Table 6 (cf. Section 4.1), carbon sensors share similarities in their performance when compared with biopolymer sensors, irrespective of differences in their hydrophilicity (cf. Table 5, Section 3.6). Despite the hydrophobic nature of activated carbon, the utility of carbon-only humidity sensors has been reported [152,153]. The application of asymmetric porous carbon films [153] was described for humidity-sensing applications. Upon exposure to an electrical current, its surface becomes oxidized by oxygen from the air, generating hydrophilic –COOH groups, which enable proton hopping via water bridges, as shown in Figure 15.

The sensing mechanism in Figure 15 correlates well with proton hopping in PANI/CHT chains (cf. Figure 9) and the flow of water in hydrophobic channels (cf. Figure 16), which will be discussed in the following section.

## 5. Experimental Strategies for the Study of Hydration

It can be inferred from the previous discussion that hydration phenomena play a key role in the structure–function properties of moisture-responsive materials, because the adsorption of water activates labile OH^−^ and H^+^ ions via the Grotthuss mechanism (cf. Equations (1)–(4), Section 1; and Figure 9, Section 3.2), which governs the sensitivity of such humidity sensors. This section provides a selective overview of hydration studies since a full treatment extends beyond the scope of this review. Instead, aspects of hydration phenomena will be covered that are relevant to the functional properties of the humidity sensor materials, as described herein for heterogeneous systems. The molecular-level details of hydration processes (solid–liquid; solid-vapor) are not particularly well understood in complex macromolecular systems, especially in the case of starch and cellulose biopolymers, as reported by Dehabadi et al. [154]. This section will discuss the challenges related to experimental studies of hydration processes using selected techniques as described above (cf. Section 1 and Table 5, Section 3.6). It can be understood that a hydrophilic biopolymer is necessary for improving the moisture-responsiveness of PANI-based sensor materials, due to the key role of hydrogen bonding in binary composites such as PANI with a biopolymer (cf. Figure 6a, Figure 11a, Figure 12 and Figure 13, *vide infra*). Hence, it is necessary to gain a molecular-level understanding of how water uptake affects the structure and properties of the PANI/biopolymer composites. In Section 3.1, the influx of water was inferred to disrupt the intermolecular PANI/CHT hydrogen bonds within the binary composite, versus the formation of H_2_O/H_2_O H-bonds. Due to its small size and versatile binding strength, water is an efficient competitor in hydrogen-bonded systems. In turn, the uptake of water by the composite may serve to disrupt the noncovalent interactions of the composite material, which results in an “*unzipping*” of the hydrogen bond network of the PANI/biopolymer complex, as illustrated in Figure 16.

Figure 15 builds upon the concept illustrated in Figure 9, where the proton-hopping mechanism occurs across the chain length of the PANI/CHT composite. As a result, the conductivity of hydrated composite will rise (cf. see also Section 3.2; and Figure 24 in Section 6). When the sample reverts to its dry state, water molecules will be desorbed, along with the reformation of intermolecular PANI/biopolymer H-bonds. The process is akin to a *ziplock*-type process, where the “*unzipped*” form corresponds to the dissociated complex and the “*zipped*” form relates to the noncovalent association complex between the polymer units (cf. Figure 16). The processing of *zipping–unzipping* will occur in a cyclic manner, along with the hydration and dehydration processes of the (bio)polymer composite. The entire cycle is referred to as hysteresis (cf. Table 6, Section 4.1), where the sensor material undergoes the maximum possible number of hysteresis cycles that determine its durability. However, the initial state will not fully recover, due to an incomplete loss of solvent, leading to the partial loss of its initial shape, which can reach ca. 20% for biopolymer-based materials (cf. Table 6). This corresponds to the scenario described in Section 3.1. (cf. Figure 6), where two possible structures of PANI/CHT composites are illustrated: hydrogen-bonded and covalent-bonded networks. Hydrogen bonding is likely responsible for the “*zip-unzip*” mechanism, whereas covalent bonding is more likely to prevent or attenuate such a process, based on a consideration of bonding in such covalent networks. The presence of partial covalent bonding may account for the role of hysteresis phenomena, since the unrecoverable loss of structure upon desorption of water that may relate to the role of covalent bonds within such types of composites. Therefore, there may be possible contributions arising from both types of structures, as illustrated in Figure 6a,b (cf. Section 3). For systems that adopt hydrogen bonding solely, this provides an account of the greater conductivity at a higher RH (“*unzipped*” form). In the case of proton-exchange membranes, the role of hydrogen bonding is well established, and is further demonstrated by the electrochemical impedance spectroscopy results at variable humidity, shown in Figure 17 [155].

Z_real_ is directly proportional to the resistivity of the sample, where the curved line at a higher RH is shifted left (lower resistivity), which corresponds to the higher conductivity of the sample. This bolsters the concept that water provides a conduit for hydrogen ions to start hopping laterally between the polymer chains or along the polymer chain length (cf. Figure 9).

Depending on the nature of the interactions between two hydrophilic polymer chains, the resulting structure may combine to yield a composite with an even more hydrophilic character. However, in the case of a composite that results in effective hydrogen bonding between subunits, the complex may result in the formation of hydrophobic domains, such as cavities or channels, allowing the migration of water to flow that supports transport of the charge carriers. This general phenomenon can be understood based on “hydrophobic effects”. The latter is distinguished from hydrophobic interactions, and is discussed further below. A detailed understanding of the kinetics and thermodynamics of hydration is required to address the aforementioned knowledge gap. Blokzijl et al. [156] introduced the term “hydrophobic hydration”, which implies that hydrophobicity and hydrophilicity may work in opposition but are closely interconnected, accounting for distinguishable processes related to hydration.

Hydrophobicity provides information about the limited solubility of nonpolar solutes in water and their subsequent aggregation. Thermodynamically, hydrophobicity is defined on the basis of the energetics of transfer from nonpolar compounds, from an apolar solvent to a water environment [156]. However, the fundamental basis of the term “hydrophobicity” has a variable interpretation, as outlined in a recent study on hydrophobic effects [157] that categorizes such phenomena into “classical” (entropy-driven) and “non-classical” (enthalpy-driven) processes. A study on cyclodextrin (CDX) hydration showed that enthalpic contributions dominate the entropy gain by the water molecules [158]. Cyclodextrins are amphiphilic oligomers of glucose, with an apolar cavity and hydrophilic functional groups external to the cavity that resemble features of cellulose, starch and chitosan. Hence, PANI/biopolymer materials will likely have an enthalpic driving force upon changes in the hydration of the apolar and/or hydrophilic domains of such macromolecular systems.

The term “hydrophobic hydration” in the literature appears to be misunderstood [156,157,158,159]. In short, it refers to the assembling of water molecules around a nonpolar solute. The process of hydration is described as (1) the formation of a cavity in the bulk of water, (2) insertion of the solute in this cavity with subsequent solute-solvent interactions and (3) rearrangement of water molecules in close proximity to the nonpolar compound. The formation of apolar domains upon the interaction of PANI with biopolymer components is anticipated for hydrogen-bonded complexes between complementary (bio)polymer units (cf. Figure 6a and Figure 11a, Section 3). The apolar surface area is likely to depend on the ratio of PANI and the biopolymer components, along with the morphology adopted by the system.

Hydrophobic effects have been thoroughly studied for CDX (cyclodextrin) complexes [160,161,162,163]. Three types of CDX structures (α, β, and γ) differ by the number of glucose subunits (6, 7, and 8, respectively). Each structure contains a cavity, which increases in size in the sequence α → β → γ. The size of the apolar cavity is important, as it may occlude more or fewer H_2_O molecules. For example, when the void is too small (less than 0.9 nm) [164], it is entropically unfavorable to fill with water where the full complement of hydrogen bonding is achieved, as compared with bulk water. In contrast, when the void becomes larger, water molecules penetrate inside the cavity and form hydrogen-bonded clusters. These clusters are more stable when they form a hydrogen-bonded network. If the water molecules escape from the cavity, the optimal interactions can be restored, where the driving force is called “high-energy water” [157]. This scenario is likely realized in medium-sized cavities (1.0 to 1.1 nm), where the driving force (referred to as enthalpic force) is directed toward the external environment. Although hydrophobic interactions seem to be well studied in the literature, their mechanisms are not completely resolved. Given the uncertainty on hydrophobic mechanisms, in PANI/biopolymer systems they appear even more complex, in view of the complex polymer structure of such composite materials.

It follows that the concept of hydrophobicity for PANI/biopolymer composites for humidity sensor materials is important, due to the role of cavities and channels, similar to that described for cyclodextrins by Blokzijl et al. [156]. Based on Figure 16, the presence of hydrophobic channels provides a conduit for water that prompts the water molecules to assemble and move as a whole. In other words, water has higher kinetic mobility (“dynamic water”), whereas hydrogen-bonded water is more static and serves as a poor conductor. Dynamic water (synonymous with the abovementioned “high-energy water”) exists in a different thermodynamic state that tends toward escape from the cavity. This is absolutely necessary to make the sample electrically conductive. To sum up, two hydrophilic polymers become more hydrophobic overall upon association but contribute to electrical conductivity. This controversial statement highlights the complexity of hydration phenomena, warranting further study to address this knowledge gap.

Hydration based on solute-water interactions can be studied via practical and theoretical approaches. A practical approach includes thermodynamic methods, such as calorimetry, along with spectroscopic characterization (e.g., NMR, Raman, XRD, SAXS etc.). NMR diffusion experiments in the solid state represent a *state-of-the-art* method for studying hydration processes. In the work reported by Thiessen et al. (2018) [165], hydration was studied from the perspective of kinetics, using double- and zero-quantum-filtered (DQF/ZQF) deuterium NMR. The study was directed at evaluating proton migration and detecting hydrophilic pockets in Nafion membranes. T_1_ and T_2_ relaxation times were calculated for trapped water as it migrated through these membranes (Figure 18). The equations that describe the line-shapes in Figure 18 using relaxation times can be found in [165] (cf. Equations (1) and (2) in [165]).

Differential scanning calorimetry (DSC) allows for estimating the enthalpy of the hydration/dehydration processes of composites. As mentioned above, enthalpy may supersede entropy as the thermodynamic driving force. Thus, the determination of enthalpy in such processes can be quite meaningful for studying hydration effects. Calorimetric studies allow for the determination of the heat flow at certain temperatures (thermal events), where each thermal event bears enthalpic information for the state of hydration of the composite. In turn, the estimation of the energetics in molar (kJ/mol) or specific (J/g) quantities can be estimated, along with the coordination number of water molecules. Water molecules adsorb onto PANI or CHT, according to the number of adsorption sites borne by each monomer unit. For example, each monomer unit of PANI possesses a single adsorption site (NH^+^, or acidic nitrogen) [166], whereas CHT may have several types of adsorption sites (-OH and -NH groups and polymer-chain end groups) [167]. This will allow for an estimate of the amount of water associated with each PANI/composite moiety, and how it depends on the relative composition of the precursors in the composite.

Interestingly, water in PANI can be detected even after being submitted to a drying cycle that relates to the so-called “fixed” and “mobile” states of water [166]. Upon drying a PANI composite at a temperature range from 30 to 300 °C for a known duration (15 min), DSC confirmed that the first thermal event was slightly higher (ca. 100 °C), indicating evaporative water loss [168]. In addition to hydrogen bonding and other non-covalent interactions, such as electrostatic forces or hydrophobic effects upon hydration, can be proven by characterization of the enthalpies of transition in calorimetric studies.

The DSC curves of PANI/CHT/PVA (PVA-polyvinyl alcohol) composites were described by Anisimov et al. [77]. Evidence of unique hydration in micropore domains and occluded water in composites was shown by thermal events that extended up to 200 °C, where an increase in the chitosan content revealed a growth in the level of incorporated water, which supports the assumption of an increase in the number of hydrophilic adsorption sites.

As shown in Figure 19, the SEM results revealed the morphological features of hydrated polyanionic cellulose composites. The extent of hydration inevitably leads to a change in the morphology [169].

Figure 19 shows that when the composite sample is exposed to moisture for a longer period of time, more regular network structures emerge. The initial porous structure of calcium-silicate-hydrate cement fills with polymer particles of polyanionic cellulose, and finally, these particles form a fusion membrane that causes a regular network structure together with the hydrated products [169]. Another work [77] revealed that PANI/CHT/PVA composites exhibited exponential growth in their adsorption capacities with respect to increased exposure to moisture.

Small-angle X-ray scattering (SAXS) is a suitable technique to study the variation in crystallinity as a function of the hydration state. For example, potato starch particles before and after hydration were studied using SAXS, where changes in the crystallinity were supported by plotting the scattered intensity *I* versus the magnitude of the scattering vector, *q* (cf. Figure 20) [170].

The SAXS signature of dry potato starch (Figure 20a) is ascribed to the lamellar semi-crystalline structure of starch [170]. After hydration, the lamellar peak became more intense and underwent a slight shift to the left, from 0.09 to 0.07 Å^−1^. This occurred due to the change from a glassy nematic state to a smectic state of starch [171]. Starch helices typically form starch crystallites; however, the alignment of starch units during hydration may change the crystalline starch types. For further understanding of how these crystalline types change, wide-angle X-ray scattering (WAXS) is required [170]. Along with the determination of crystalline structure, SAXS can also aid in determining the attractive or repulsive interactions between particles. For instance, electrostatic/non-electrostatic interactions can be calculated through the structure factor (S), which is derived from scattered intensity (I). Such types of interactions are significant for understanding the structure-function properties in PANI-based composites and may also give a clue about their humidity-sensing mechanism. When a hydrogen-bonded composite is formed, it becomes more ordered and crystalline. After the infiltration of water, it becomes less crystalline. This yields a dynamic system, with higher conductivity in its amorphous (wet) state and lower conductivity in its crystalline (dry) state.

The molecular-level details concerning the nature and mechanism of the hydration of PANI-based materials are not fully understood and remain an area that warrants further research via complementary experimental approaches. Due to challenges related to interpreting molecular-level details from experiments, computational studies may serve to provide insight into the nature of the hydration processes, energetics and kinetics of the system. Theoretical approaches that are based on molecular dynamics and computational methods can be used to gain insight into the hydration properties of biopolymers during the adsorption process [172]. Computational studies have been carried out for biopolymer/dye systems that provide valuable insight into the role of hydration during adsorption processes (cf. Figure 5 from [172]).

Water absorption by PANI is a dual process that occurs within the internal microstructure and outer surface sites of the composite. Water molecules may be bound to the acidic sites of PANI via hydrogen bonding, where a proposed model [173] illustrates that PANI is surrounded by several hydration shells that include 5, 10, and 20 molecules of water around a single unit of PANI (cf. Figure 21). The respective hydration shells were simulated with the aid of density functional theory (DFT) calculations and the B3LYP/3-21G** model. The results showed the build-up of water at the NH-groups of PANI that yield enhanced conductivity.

Figure 21 schematically shows the water molecules that accumulate at the hydrophilic sites of PANI. The model indicates that four H_2_O molecules are attached to the NH-groups of PANI, and one molecule for the H-atom at the meta-position of a benzenoid ring of PANI. These five molecules form the first hydration shell. The second shell is formed by 10 molecules of water, which occur via H-bonding to water in the first shell. The third shell accumulates 20 H_2_O molecules, each of which is bound to the H_2_O of the previous shell. An illustration of the original ball-and-stick model obtained by DFT calculations was reported by Alghunaim in 2019 (cf. Figures 3–5 in [173]).

To gain further insight on the role of hydration processes in humidity sensor materials, a wide range of complementary experimental and computational methods can be used, as outlined in the selected examples above. A selected overview of recent studies that focus on hydration-related studies in diverse types of chemical systems is summarized in Table 7.

## 6. Material Design Approach for Unique Hydration Properties

PANI is an ideal polymer matrix for materials with unique hydration properties. Several factors should be considered in the design of a suitable humidity sensor device at each level of fabrication, such as the sensitive layer, electrodes, substrate layer, etc. The first element is the sensitive layer, which is responsive toward moisture. PANI itself has a limited hygroscopic character and moderate hydrophilicity; however, its electrical conductivity depends on the extent of hydration. For a PANI-based composite to be a good humidity-sensing material it may be modified with a secondary material in order to enhance its hydration properties. Additional blending with a third component will afford the desired property of being sufficiently hygroscopic, as reported by Anju et al. [12].

In addition to adding other polymer-based materials to PANI to improve its hydration properties, PANI can be functionalized with different chemical groups. The humidity-sensing properties can be precisely tuned, depending on the functional groups introduced. These functional groups can be either hydrophilic or hydrophobic. Lyuleeva et al. [181] proposed that hydrophilic methacrylic acid (MAA) and hydrophobic *tert*-butyl acid (*t*-BMA) functional groups can be grafted onto a moisture-responsive material, as a way of tuning the humidity-sensing performance of the materials.

Among the multitude of functionalized polyanilines are halogenated PANI [182,183], carboxylic, sulfonic and sulfamic derivatives of PANI [184], and butylthio-functionalized PANI [185]. Both the protonated and non-protonated forms of PANI will undergo changes in band gaps, highest occupied molecular orbital (HOMO), and lowest unoccupied molecular orbital (LUMO). Therefore, its chemical and electronic properties will be altered, where such alteration can be precisely tailored to the required humidity performance of a sensor. The polarity and size of the functional group are the primary criteria that impart sensitive properties through appropriate functionalization. However, these changes will not significantly influence the intrinsic properties of PANI, such as conductivity [186]. This allows for the tuning of PANI to an adequate level of hydrophilicity, while maintaining its conductive properties. A hydrophobic material with good porosity can also adsorb water, examples of which include cyclodextrins and porous activated carbon (cf. carbon aerogel from Section 4.1).

The second element is the lower layer located beneath the sensitive layer. This should be a substrate, typically made of glass, alumina, or silicone since they are insulators [187], along with embedded electrodes (cf. Figure 22).

The third important component is the electrodes, which can be made from ultrathin metal plates. In general, copper, nickel, gold, or aluminum serve as good electrical conductors for electrodes. The ultimate humidity sensor can be fabricated according to a resistive or capacitive mode [188] since the humidity response can be measured via either resistance [10] or capacitance [12]. The modality of measurement depends on several factors: (1) the content of PANI; (2) the range of measurable relative humidity; and (3) the temperature of the environment.

PANI/biopolymer films should serve as a promising component for resistive-type humidity sensors since their change in resistance as a function of RH can be precisely measured. The preparation of samples in the form of rectangular or circular shapes depends on the type of resistivity measurements, where two modalities are often employed: two-terminal (2T) and four-terminal (4T) sensing (cf. Figure 23).

The two-point sensing technique is the simplest and more established method to measure electrical resistivity, one that is based on the voltage drop and current measurement across a sample (Figure 23a). It works well when the sample has high resistivity (e.g., low conductivity, as reported for PANI/biopolymers [189] (cf. Table 9 also for additional examples, *vide supra*). The net equation for resistivity (ρ) is [190]:(10)$$\rho =\frac{US}{lI}\;$$

*S* and *l* are the cross-sectional area and length of a conductor, respectively. The ratio *U/I* is measured as a slope on the current-voltage graph. When the conductor behaves as a semiconductor, the trend for *U/I* may differ from a straight-line relationship, where some corrections may be required [77].

The four-point sensing technique (also known as a 4T Wenner probe [191]) is used to conduct more accurate measurements, especially for the evaluation of the sheet resistivity of thin films [192]. This method is based on four electrodes, each pair of which bears current (*A*) and voltage (*V*) separately (Figure 23b). This method is also convenient since the resistivity of regular circular samples does not depend on the diameter (*d*), but only on thickness (*h*); the resistivity can be calculated, if resistance *R* is known, by Equation (11):(11)ρ=Rh 

This method is applicable for samples with moderate and low resistivity and is especially useful for samples that experience a sudden drop in resistivity or a boost in conductivity under high RH [190,193]. It is clearly seen that resistivity depends on the thickness of films, which in turn affects the response time of a sensor. Packirisamy et al. [188] fabricated a polyimide-based resistive humidity sensor and showed the difference between two samples of variable thickness (cf. Figure 8 in [188]).

The response curves were fitted according to Equation (12):(12)$$R=\frac{C}{RH-RH_{c}}+R_{0}\;$$
where *R* is the resistance in MOhm, *RH* is relative humidity (%), and *RH_c_* is the so-called “cut-off” humidity, which indicates a lower RH limit, after which the sensor behaves as an insulator; *C* is a fitting constant, and *R*_0_ is residual resistance due to impurities.

To test the sample as a humidity sensor, it is placed into a chamber according to Figure 24. The chamber is connected to a hygrometer in order to measure RH inside the compartment, and to a potentiostat, denoted as “Keithley” in Figure 24. The vacuum pump ensures an air and water vapor outlet exhaust if needed. The vaporizer nebulizes moisture into a chamber, which can be replaced by a saturated salt solution for simplicity, as described in a recent review by Singh and Shukla [59] (cf. Figure 12 in [59]). A fan ensures the uniform distribution of water vapor around the chamber. The change in conductivity/resistivity at different RH values is measured with a potentiostat.

It is understood that the cut-off humidity constitutes 41.8% for polyimide sensors (cf. Figure 7 in [188]), which implies that the lowest operational RH for such samples is near 45%. The practical working range of the RH is an important feature of humidity sensor materials, as described above. To optimize the desired humidity response and to fabricate the appropriate moisture-sensitive material, a central composite design [195] and Box–Behnken methodology [196] offer a reliable statistical approach. These methods are designated for multifactorial optimization; however, the Box-Behnken design (BBD) is preferred when the number of variables is three or greater [197]. BBD allows for calculating the minimum number of experiments *(N)* that are needed for such an optimization, where N can be calculated by Equation (13) [198]:*N* = 2*k*(*k* − 1) + *C_o_*(13)
where *k* is a number of factors (3 in the given case) and *C_o_* is the number of central points (equals to 1). Consequently, to optimize a humidity response, 13 experiments are required. Each experiment is denoted as a point in the middle of a cube edge, and the last point is at the center of the cube (cf. Figure 25) [197].

The advantage of the Box–Behnken design is that it enables the selection of the optimal conditions of an experiment, excluding extreme cases, which results in the prevention of failed experiments. Each level of condition is coded as –1, 0 and 1, which can be interpreted as “low”, “moderate” and “high” values, respectively. Table 8 displays such conditions, if it is assumed that the humidity-responsive material would operate approximately at room temperature.

All possible points for the BBD (12 points + C—central point), avoiding extrema conditions, are shown in detail elsewhere (cf. Table 1 in [197]).

The BBD was employed successfully for various types of polymer-based sensors’ design [199], optimization of mechanical properties [200], and the study of electrochemical processes [201] among various other examples. The great advantage of BBD is that it enables the alteration of multiple factors simultaneously and the prediction of a resulting response for such changes. In contrast to that, the one-factor-at-a-time (OFAT) approach [202] is slow and time-consuming. Another advantage is that BBD will aid in the determination of the optimal conditions of the experiment with high precision, which may not be achieved by the regular OFAT method.

## 7. Discussion and Knowledge Gaps

Although the coverage of the scientific literature reported for various humidity-sensing materials appears to be broad in scope, there are limited reports on PANI/biopolymer composites as potential humidity sensors. Various review articles [30,64] indicate that the humidity-sensing performance of PANI-based composites have received good coverage for PANI in combination with metal additives (e.g., Co, Cd, Ag), metal oxides (e.g., TiO_2_, WO_3_, V_2_O_5_) and metal chalcogenides (e.g., CdS, CdSe). Some of these metals that have been developed as sensors may suffer from limitations due to material cost and the relative availability of materials, in some cases. Therefore, there is a need to find alternatives that are more cost-effective and sustainable, such as biopolymers and oligomers: chitin, chitosan, cellulose, starch, cyclodextrin, or carrageenan. Table 9 provides a summary of relevant data for PANI/biopolymer humidity-sensing materials. The results as shown reveal that such materials possess features such as porosity, moisture absorption, electrical conductivity, and response-recovery times, which are among the most important design considerations for humidity sensor devices. However, many entries in this table remain unreported, which reveals the need for further study in this field.

For instance, the electrical conductivity of many PANI/biopolymer samples is known, whereas the PANI/cellulose samples have the highest conductivity among the various biopolymer composites, as compared with those containing chitosan and chitin. It may appear that chitin/chitosan is not suitable for humidity sensing. However, the conductivity of the entire composite depends on the component ratio, and on the presence of other (additive) components, including the degree of PANI-doping. A knowledge gap in the role of multi-factorial effects remains to be studied, which can be addressed through statistical optimization using methods such as the Box–Behnken design.

There are knowledge gaps for other relevant humidity-sensing parameters. For example, the porosity for the PANI/biopolymer composites has not been widely studied, especially under relevant environmental conditions. In the case of studies other than SEM imaging, an estimate of the porosity (microporous, mesoporous, and macroporous) of such materials by this approach is prone to uncertainty, particularly for samples in an anhydrous state that are likely to undergo solvent swelling. The surface accessibility of pores and the average pore diameter can be estimated from the Brunauer-Emmett-Teller (BET) analysis of vapor adsorption isotherms. In contrast to rigid carbonaceous or inorganic solids, biopolymers can undergo incremental swelling, with greater levels of water uptake (vide supra). This is in contrast to ceramic materials that are rigid in nature, and their structure is relatively stable regardless of the moisture content or RH. Hysteresis in a sensor device is also important, as discussed in Section 4.1. Although biopolymers usually have greater hysteresis effects relative to ceramic materials, the performance still resides within an acceptable range.

To design a good humidity sensor, the response and recovery times are key parameters that have not been widely studied. Moisture absorption, calculated as a percentage increase by mass, represents a potential topic area with limited reported research. In the case of the design of humidity sensor materials, it is a key feature that governs the sensitivity and efficiency parameters. Finally, as PANI composites are relatively new materials for use as humidity sensors, the literature reviewed herein highlights the need for further studies on humidity sensor design and their functional properties. In particular, this statement refers to the use of soft (biopolymers) versus hard (ceramic materials), in conjunction with PANI.

## 8. Summary and Future Outlook

In conclusion, PANI is a preferred candidate for humidity sensor materials with unique hydration properties. It has an electrical conducting mechanism that relies on the role of molecular interactions between water and the hydrophilic sites of PANI. The conductivity of PANI is highly dependent on the ambient RH, which is a key factor to consider in the fabrication of such devices. To enhance the hydrophilic properties of PANI, it can be modified by the formation of composites with biopolymers, such as chitosan, cellulose, starch, cyclodextrin and other types of polysaccharides that facilitate the hydrogen-bonded composites. The formation of these composites ultimately affect the structure and surface chemical properties of the system. The moisture-responsive mechanism of PANI/biopolymer systems depend on two factors: (i) electron/proton hopping between water molecules and acidic nitrogen groups of PANI; and (ii) on the morphology of the composites, which can be altered under the influence of relative humidity (curling–uncurling of PANI/biopolymer chains). The kinetics of this process have not been fully reported to date, despite the importance of kinetic factors that affect the performance of fast-response humidity sensor devices. In practice, the hydration properties of PANI/biopolymer materials can be investigated with the aid of complementary methods that provide insight into the thermodynamics, kinetics, and structural aspects of the humidity-sensing process. Computational methods are anticipated to provide further insight on the structure of hydrated PANI-based composites, through the deployment of methods such as molecular dynamics and DFT calculations (cf. Table 7).

PANI-based composites are normally obtained via oxidative polymerization in acidic media. The nature of the dopant (acid) plays an important role as a template in the resulting morphology and humidity-sensing performance of the composites. A secondary component (biopolymer) results in a synergistic effect for improving the sensitivity and performance of PANI but may affect the electronic properties (conductivity) of PANI, according to structural considerations. The conductivity of PANI is an intrinsic property that is influenced by the functionalization of PANI via physical or chemical means.

The configuration of a humidity sensor consists of at least two components: (i) a sensitive layer that may comprise a PANI-based composite, and (ii) a substrate layer with embedded electrodes, where the humidity response is measured as impedance (resistance) or capacitance at variable RH. To optimize the humidity-sensing performance, multifactorial design approaches have been employed, such as the Box–Behnken design. The latter is desirable and serves to guide experimental design for the optimization of multi-parameter systems.

Nevertheless, the literature on PANI-based moisture-responsive materials is at an early stage, in comparison with conventional ceramic and carbonaceous sensor materials. A perspective drawn from this contribution recommends that further studies be directed at gaining an improved understanding of humidity-sensing mechanisms for PANI/biopolymer-based nanocomposites. Dedicated experimental and computational studies are anticipated to achieve this goal, through the use of complementary experimental methods. The holistic approaches and methodologies outlined herein are anticipated to provide a molecular-level understanding of the kinetics and thermodynamics of hydration for such systems. In turn, emerging materials such as PANI-based biopolymer nanocomposites are envisioned to occupy an increasingly important role in the future development of sensor technology that employs PANI-based materials for diverse applications.

## Figures and Tables

**Figure 1 polymers-13-02722-f001:**
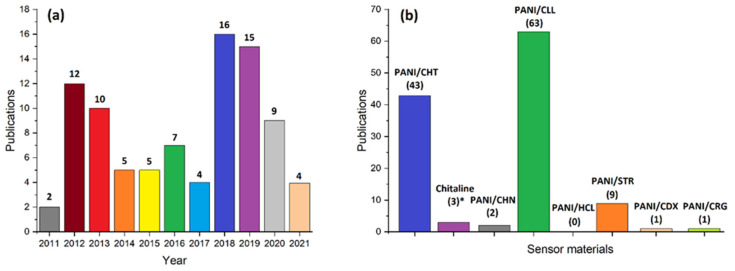
Statistical survey on recent publications: (**a**) polyaniline-based humidity sensors in the last decade; (**b**) total publications to date for polyaniline-biopolymer composites (according to the search queries stated in the graph). Source of data: Scopus; search time: 2 July 2021. Acronyms: CHT—chitosan, CHN—chitin, CLL—cellulose, HCL—hemicellulose, STR—starch, CDX—cyclodextrin, CRG—carrageenan. Chitaline was searched by a plain request “chitaline”.

**Figure 2 polymers-13-02722-f002:**
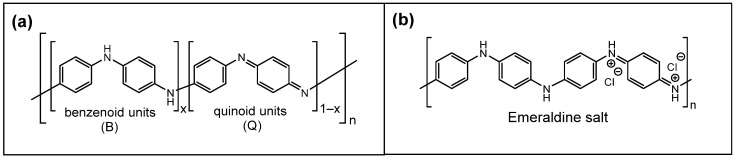
Structures of PANI: (**a**) general structure showing benzenoid (B) and quinoid (Q) units in its base form; and (**b**) emeraldine salt (PANI-ES). Adapted from [64,65]. The x-value is defined as the fraction of benzenoid (B) and quinoid (Q) units, as follows: -(B)_x_-(Q)_1−x_- for a given degree of polymerization (n), according to panel (**a**).

**Figure 3 polymers-13-02722-f003:**
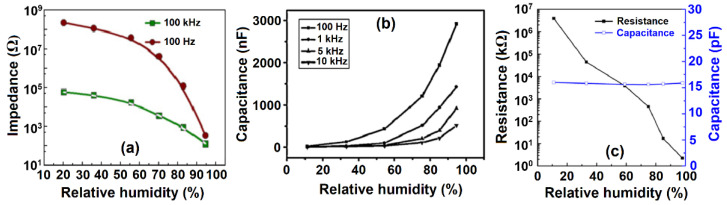
Panel (**a**): impedance (real) vs. RH dependence for a ZnO-based resistive humidity sensor. Reprinted with permission from [79]. Copyright 2013, Elsevier. Panel (**b**): capacitance vs. RH curve for ZnO-based capacitive humidity sensor. Reprinted with permission from [80]. Copyright 2013, Elsevier. Panel (**c**): comparative resistance and capacitance vs. RH curves for a mesoporous resistive LaFeO_3_ sensor. Reprinted with permission from [81]. Copyright 2013, Elsevier.

**Figure 4 polymers-13-02722-f004:**
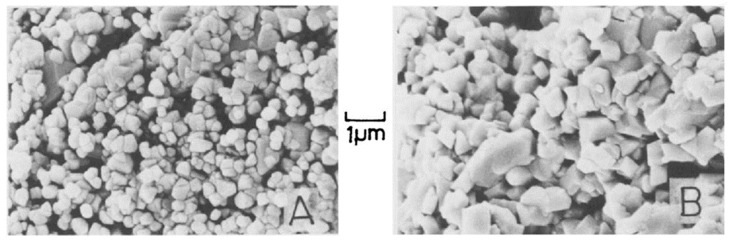
SEM images of porous ceramics: (**A**) MgCr_2_O_4_; (**B**) 7MgCr_2_O_4_·3TiO_2_. Reprinted with permission from [83]. Copyright 1980, The American Ceramic Society.

**Figure 5 polymers-13-02722-f005:**
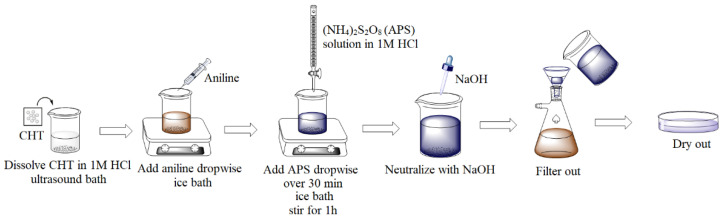
Generalized synthetic procedure for PANI/CHT binary composites.

**Figure 6 polymers-13-02722-f006:**
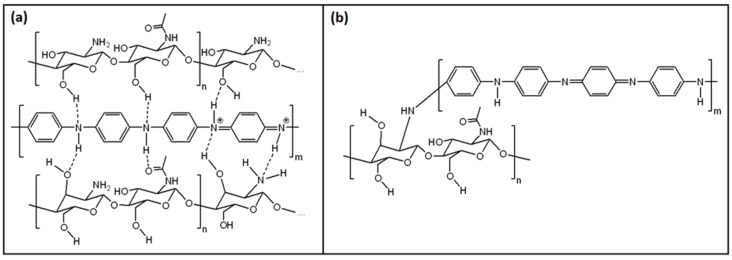
Schematic structures of chemical bonding between PANI and CHT units: (**a**) hydrogen bonding (PANI/CHT); (**b**) covalent bonding via grafting (PANI-g-CHT; “chitaline”). Adapted with permission from the results presented in [77,90,91,92,93,94,95,96,97].

**Figure 7 polymers-13-02722-f007:**
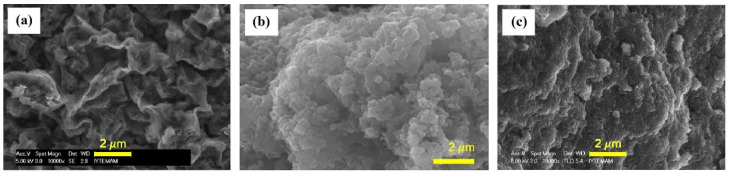
SEM images of PANI/CHT composites (in their powdered form) that were synthesized in various acid media: (**a**) CH_3_COOH, (**b**) HCl, (**c**) H_2_SO_4_ medium. Panels (**a**,**c**) reprinted with permission from [84]. Copyright 2011, Elsevier. Panel (**b**) reprinted with permission from [85]. Copyright 2018, Wiley Online Library.

**Figure 8 polymers-13-02722-f008:**
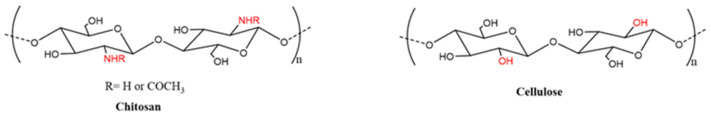
Chemical structures of chitosan and cellulose. Reprinted with permission from Ref. [99]. Copyright 2021, MDPI.

**Figure 9 polymers-13-02722-f009:**
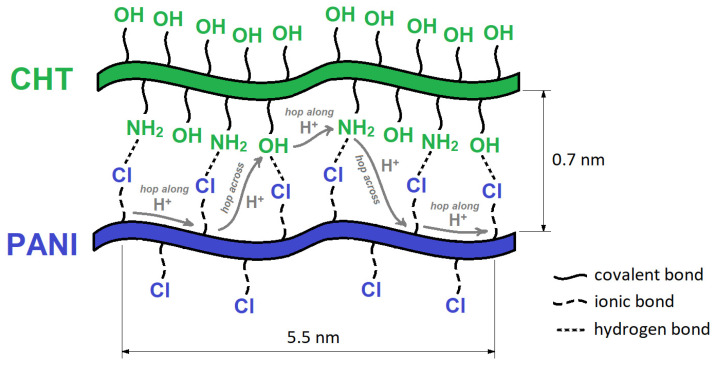
Illustrative view of the proton-hopping mechanism in PANI/CHT composites. The estimated dimensions of PANI/CHT chains are based on the covalent radii of the atoms.

**Figure 10 polymers-13-02722-f010:**
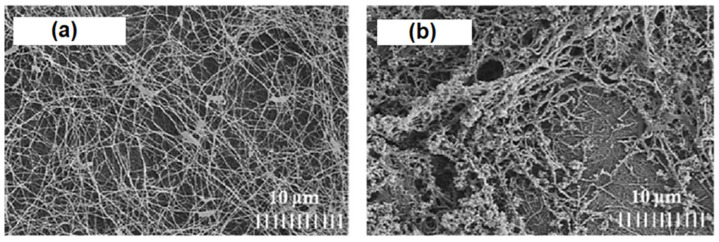
SEM micrograph of (**a**) bacterial cellulose; (**b**) PANI/BC composite. Reprinted with permission from [129]. Copyright 2018, Elsevier.

**Figure 11 polymers-13-02722-f011:**
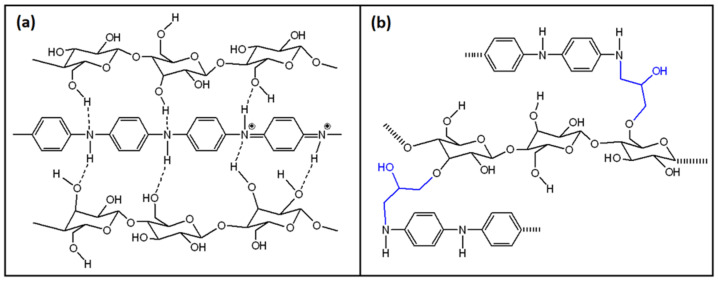
Schematic representation of polymer structures: (**a**) PANI/BC composite; (**b**) PANI/*g*EBC composite (cross-linker is highlighted in blue). Based on the results reported in [114,118,119,126]. EBC—epoxy-modified bacterial cellulose.

**Figure 12 polymers-13-02722-f012:**
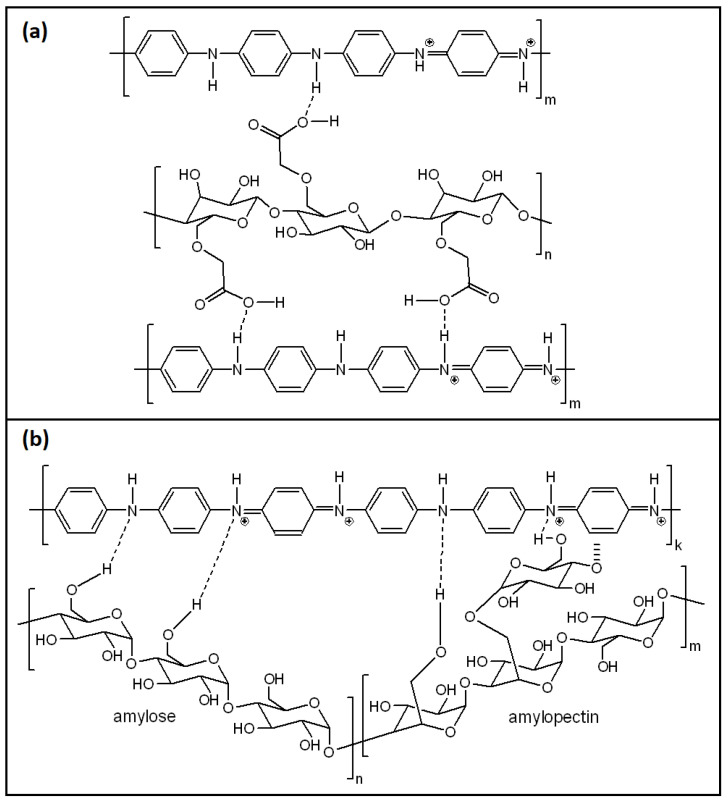
Schematic representation of polymer structures: (**a**) PANI/CMC composite; and (**b**) PANI/STR composite. The terms “m” and “n” refer to the degree of polymerization for the amylopectin and amylose segments, respectively. Based on the results from [124,139].

**Figure 13 polymers-13-02722-f013:**
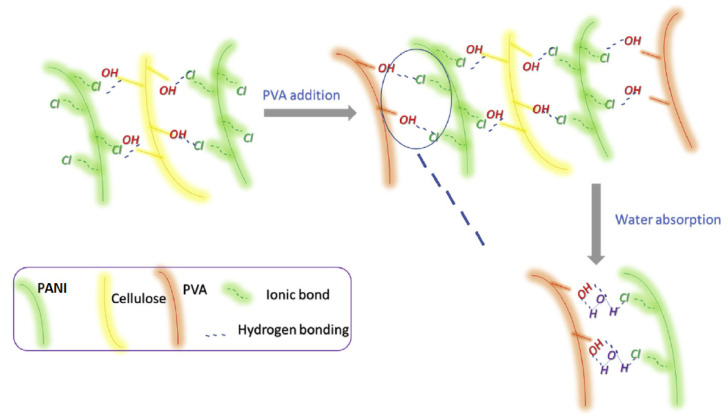
An illustration of the conceptualized structure of a PANI/NFC/PVA composite, to account for its humidity sensitivity. Reprinted with permission from [12]. Copyright 2019, Elsevier.

**Figure 14 polymers-13-02722-f014:**
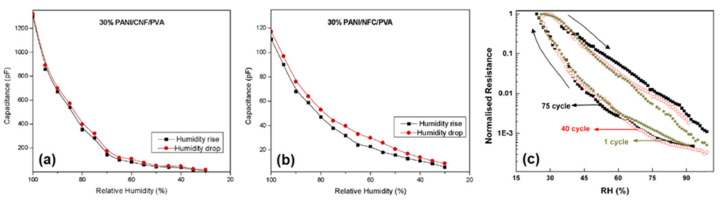
Hysteresis response curves for various composites: (**a**) PANI/CNF/PVA; (**b**) PANI/NFC/PVA; and (**c**) PANI/CHN. Panels (**a**,**b**) reprinted with permission from [12]. Copyright 2019, Elsevier. Panel (**c**) reprinted with permission from [10]. Copyright 2010, Elsevier.

**Figure 15 polymers-13-02722-f015:**
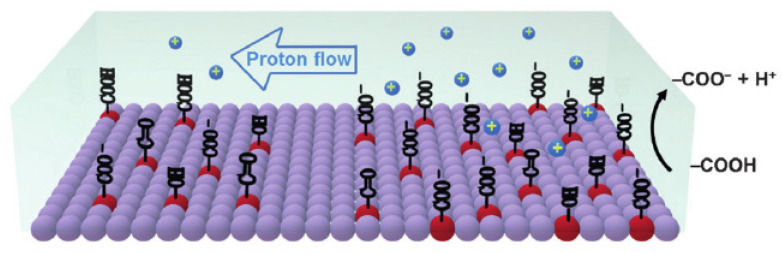
The proton-transfer mechanism in functionalized porous carbon film. Reprinted with permission from [153]. Copyright 2016, Wiley Online Library.

**Figure 16 polymers-13-02722-f016:**
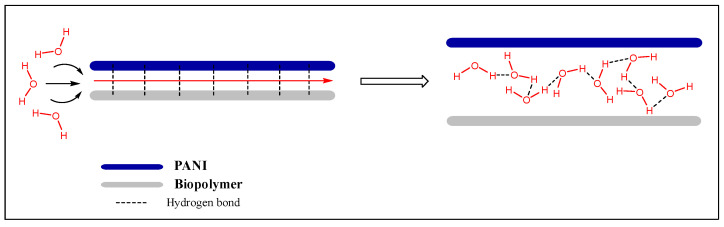
A conceptual structural model of a hydrogen-bonded composite in an anhydrous form (initial state; **left**) and accompanying dissociation of the polymer complex (final state; **right**) at elevated relative humidity levels.

**Figure 17 polymers-13-02722-f017:**
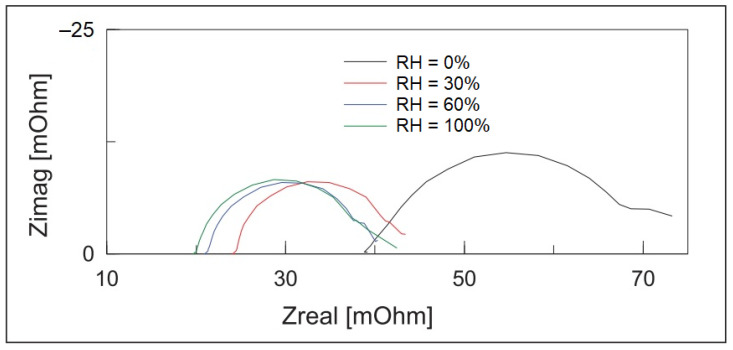
Nyquist plot (imaginary vs. real impedance) derived from electrochemical impedance spectroscopy (EIS) for a proton-exchange membrane in a fuel cell. Reprinted with permission from [155]. Copyright 2012, Elsevier.

**Figure 18 polymers-13-02722-f018:**
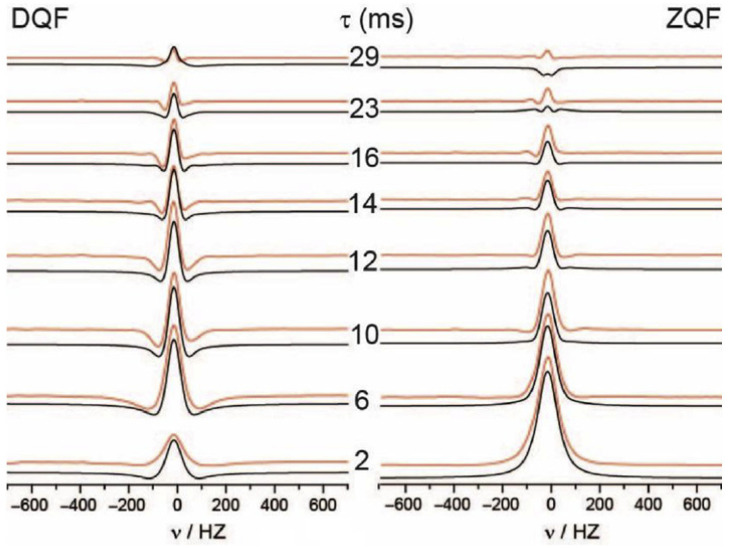
Deuterium DQF/ZQF NMR spectra of a Nafion membrane, hydrated to 15 D_2_O molecules per sulfonate group. Reprinted with permission from [165]. Copyright 2018, Elsevier.

**Figure 19 polymers-13-02722-f019:**
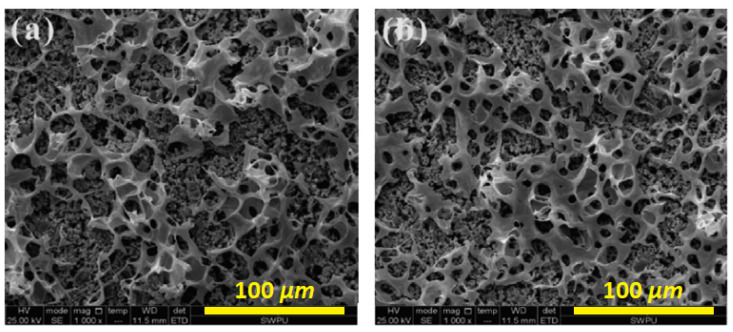
SEM images of the polyanionic cellulose-reinforced cement, with different degrees of hydration: (**a**) smaller; and (**b**) larger. Reprinted with permission from [169]. Copyright 2020, Elsevier.

**Figure 20 polymers-13-02722-f020:**
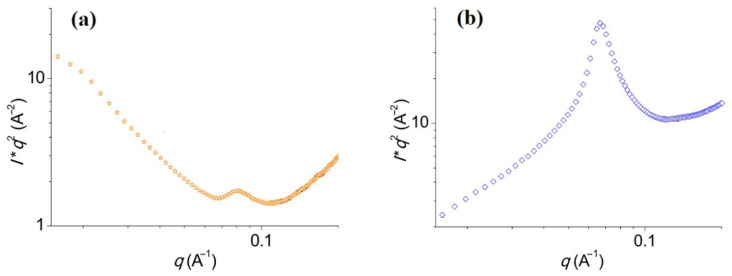
Lorentz-corrected synchrotron SAXS patterns of potato starch particles: (**a**) before hydration; and (**b**) after hydration. Adapted with permission from [170]. Copyright 2017, Elsevier.

**Figure 21 polymers-13-02722-f021:**
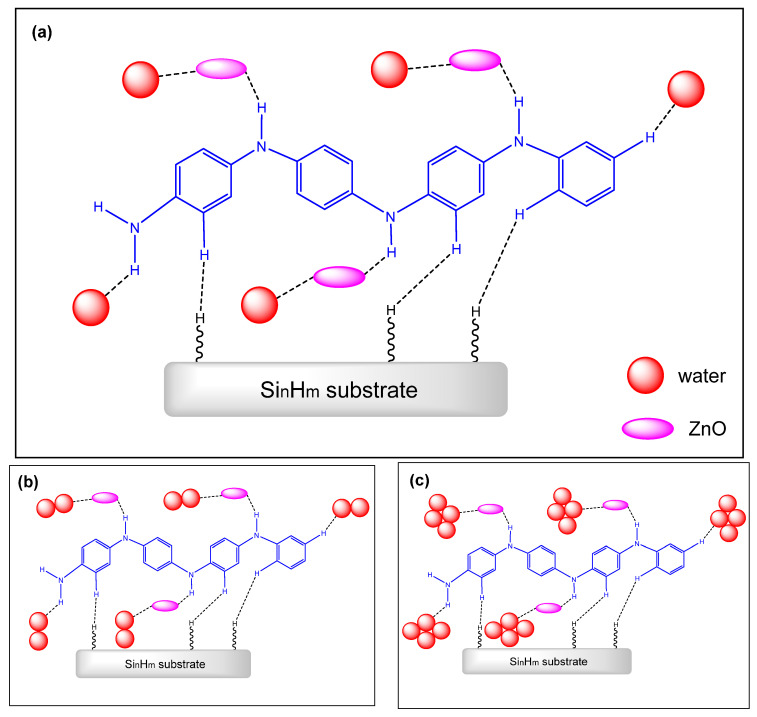
B3LYP/3-21G** calculated structural model for a hydrated PANI-Si-ZnO composite: (**a**) first, (**b**) second, and (**c**) third hydration shells of the PANI-Si-ZnO composites with 5, 10, and 20 molecules, respectively. Based on DFT results from [173].

**Figure 22 polymers-13-02722-f022:**
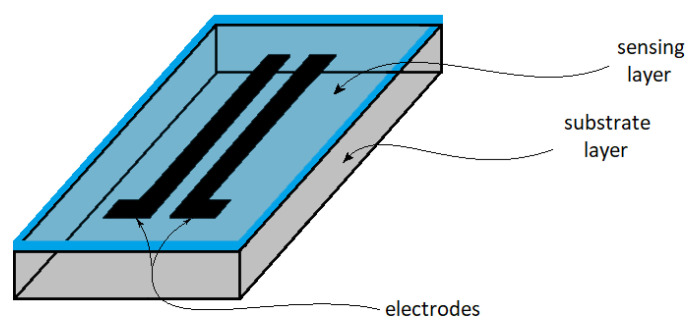
Simplified sensor configuration, with the substrate, electrodes, and sensing layer above them.

**Figure 23 polymers-13-02722-f023:**
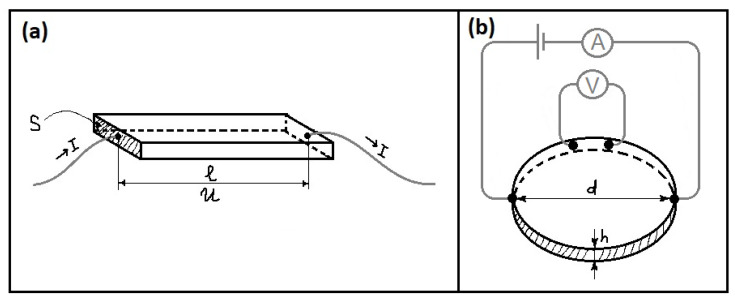
Schematic representation of two types of resistivity measurements: (**a**) two-point (2T) probe; and (**b**) four-point (4T) probe methods.

**Figure 24 polymers-13-02722-f024:**
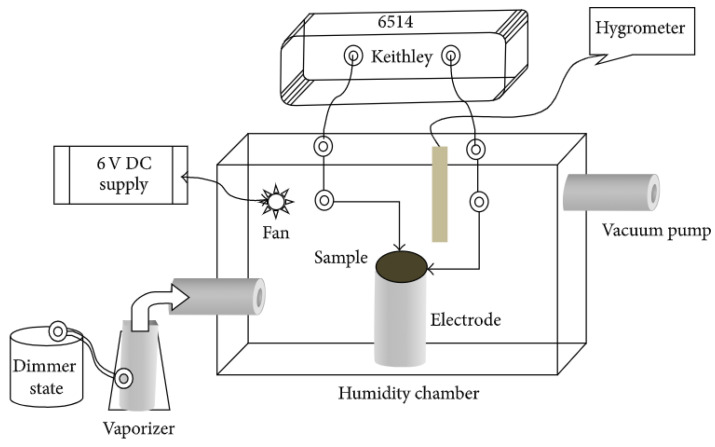
Schematic sketch of a humidity-sensing setup. Reprinted with permission from [194]. Copyright 2014, Hindawi.

**Figure 25 polymers-13-02722-f025:**
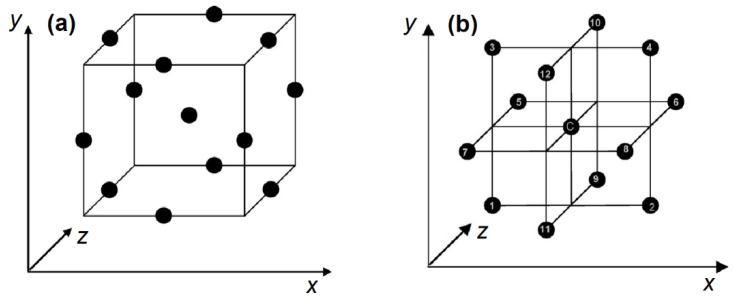
Two different representations for a 3-factorial Box-Behnken design: (**a**) a cube; and (**b**) a three interlocking 2^2^ factorial design. Reprinted with permission from [197]. Copyright 2007, Elsevier.

**Table 1 polymers-13-02722-t001:** Common humidity sensor types: resistive, capacitive, and thermal [29].

	Type	Resistive	Capacitive	Thermal
Properties				
**Basis**		Conducting moisture-absorbing film	Two conducting plates with a hygroscopic dielectric located in between	Two thermal sensors: one encased in dry air (or N_2_), where another is exposed to humid air
**Measurand**		Resistance	Capacitance	Temperature
**RH range (%)**		5–90	Full range (0–100)	Measures the absolute humidity in g/m^3^
**Temperature range (°C)**		−40–100	−50–150	90–300
**Accuracy**		Moderate	High	Depends on the temperature, where a value above 90 °C is optimal
**Price**		Low to moderate	High	Varies
**Potential applications**		Smart food packaging, automotive devices, residential environments	Medicine, scientific research	Industry, machinery; clothes, textile and wood drying

**Table 2 polymers-13-02722-t002:** Various types of resistive humidity sensors.

Material Type	Classification	Sub-Classification	Examples
Polymer	Polymer-electrolyte	Quaternary ammonium, sulfonate and phosphonium salts	AEPAB/styrene [35] PEDOT:PSS/GO [36] VBTPC/*n*-butylacrylate [37]
	Conducting polymers	Polyaniline, polypyrrole, poly(ortho-phenylenediamine), polythiophene	PANI/CHN [10] PPy/graphene oxide [14] PoPD/PANI [15] P3HT/Cu_3_(BTC)_2_ [16]
Ceramic	Electronic/ionic conduction mechanism	Perovskites	MnPS_3_ [1], BaTiO_3_ [2], Ba_0.5_Sr_0.5_TiO_3_ [3]
		Thick film ceramics	Gd-doped CeO_2_ [4], Fe_2_O_3_/SiO_2_ [5]
		Cation-doped ceramics	Na^+^/K^+^-doped Ga_2_O_3_ [6], La^3+^-doped BaSnO_3_ [7]
		Thin-film ceramics	Mn_1.2_Co_1.5_Ni_0.3_O_4_ thin film [8], WO_3_/TiO_2_ thin films [9]
Polymer/Ceramic	Electronic/ionic conduction mechanism	Polymer/metal	PVP/Ag [38], PVA/Ag [38], PVP/Au [39], PANI/PTFMA/Ag [40]
		Polymer/metal oxide	PVA/SnO_2_ [41], CLNF/ZnO [42], PANI/CuO [43], CHT/CuMn_2_O_4_ [44], PDMS/Armalcolite [45]
		Polymer/inorganic salt	PANI/MgCrO_4_ [46] PPy/Sr_3_(AsO_4_)_2_ [47] SPEEK/CaCl_2_ [48] PVA-PAA/NbC [49]

AEPAB—[2-(acryloyloxy)ethyl] dimethyl propyl ammonium bromide; PEDOT—poly(3,4-ethylenedioxythiophene); PSS—polystyrene sulfonate; VBTPC—(vinylbenzyl) tributyl phosphonium chloride; PPy—polypyrrole; PoPD—poly(o-phenylenediamine); P3HT—poly(3-hexylthiophene-2,5-diyl); BTC—1,3,5-benzenetricarboxylate; PVP—polyvinylpyrrolidone; PVA—polyvinyl alcohol; PTFMA—polytrifluoromethyl aniline; CLNF—cellulose nanofibers; PDMS—polydimethylsiloxane; SPEEK—sulfonated poly (ether ketone); PAA—polyacrylic acid.

**Table 3 polymers-13-02722-t003:** PANI in various forms and oxidation states *.

*x* Value	Oxidation State	Name	Base Form	Color	Salt Form	Color
1	fully reduced	leucoemeraldine	PANI-LB	colorless	PANI-LS	light yellow
0.5	half-oxidized	emeraldine **	PANI-EB	blue	PANI-ES	green
0	fully oxidized	pernigraniline	PANI-PB	violet	PANI-PS	dark blue

* Data obtained from [64,65,66,67]. ** The most stable emeraldine form of PANI is highlighted in grey for emphasis.

**Table 4 polymers-13-02722-t004:** Pore structure, resistivity, and conductivity of MgCr_2_O_4_–TiO_2_ sintered composites [83].

Specimen	Grain Size (µm)	Pore Size (nm)	Surface Area (m^2^/g)	Resistivity (Ω·cm)	Conductivity (S/cm)
MgCr_2_O_4_	0.2	100	1.6	1.3 × 10^10^	7.7 × 10^−11^
7 MgCr_2_O_4_·3TiO_2_	1	270	0.3	2.6 × 10^12^	3.8 × 10^−12^
3 MgCr_2_O_4_·7TiO_2_	4	350	0.1	1.0 × 10^9^	1.0 × 10^−9^

**Table 5 polymers-13-02722-t005:** Comparative features of PANI/biopolymer and PANI/carbon humidity sensors.

Feature	PANI/Biopolymer	PANI/Carbon
Conductivity of components	One is non-conductive	Both are conductive
Hydrophilicity	Biopolymer is hydrophilic	Carbon is hydrophobic
Type of interactions	H- or covalent bonding	van der Waals interactions
Size of a fiber	Biopolymer is thin	Carbon is thick

**Table 6 polymers-13-02722-t006:** Comparative features of carbon- and biopolymer-based sensors.

Material	Type of Sensor	Response Time (s)	Recovery Time (s)	Hysteresis Error (%)	Ref.
**PANI/carbon sensors**					
PANI/MWCNT	Resistive	60	140	0.5	[149]
PANI/CNF/PVA	Capacitive	41	50	1	[12]
PANI/CA (H_2_S)	Resistive	1	960	—	[150]
PANI/G (CO_2_)	Resistive	81	20	—	[151]
**PANI/biopolymer sensors**					
PANI/NFC/PVA	Capacitive	47	58	5	[12]
PANI/CHN	Resistive	30	180	18–20 *	[10]
PANI/CMC	Resistive	10	90	2	[13]

* Depends on the number of cycles.

**Table 7 polymers-13-02722-t007:** Selected experimental studies for the study of hydration processes.

Category	Experimental Method	Application	Ref.
**Spectroscopy**	^2^H NMR diffusion	Detection of hydrophilic pockets	[165]
	^1^H NMR	Intramolecular H-bonding by calculation of J-coupling constants	[174]
	SEM	Morphology as a function of variable hydration	[169]
	SAXS/WAXS	Crystallinity as a function of variable hydration	[170]
	NMR crystallography	Noncovalent interactions, detection of labile H atoms	[175]
	Raman spectroscopy: D_2_O spectral probe	Bound-water fraction by detection of HOD uncoupled oscillators via isotopic dilution	[154]
**Calorimetry**	DSC	Enthalpies of hydration and dehydration (solid-vapor or solid-liquid systems)	[77,168]
	ITC	Solvent binding enthalpy (solid-liquid system)	[176]
	Immersion calorimetry	Immersion enthalpies in water (liquid or vapor phase)	[177]
**Thermodynamic**	Water vapor adsorption isotherms	Surface area and pore volume at variable temperature for solid-vapor systems	[177,178]
	Equilibrium dye adsorption	Use of dye probes to estimate the hydrophile-lipophile character of adsorbent and water adsorption capacity	[77]
	GIST	Enthalpy and entropy contributions to the free energy of solvation	[179]
	Hydration distribution model	Thermodynamics of pocket hydration	[180]
**Computational**	Quantum-Mechanical DFT	Accumulation of water molecules at hydrophilic sites	[173]
		Correlation of dye-based adsorption with hydration	[172]
	Physical models	Derivation of the equations for water binding (free energies versus chemical potentials and activities)	[176]

ITC—isothermal titration calorimetry; GIST—grid inhomogeneous solvation theory.

**Table 8 polymers-13-02722-t008:** Numerically encoded conditions of the experiment.

	−1	0	1
PANI (% *w*/*w*)	25	50	75
RH, %	35	65	95
T, °C	15	25	35

**Table 9 polymers-13-02722-t009:** PANI/biopolymer humidity sensing materials and their relevant properties.

Material	Porosity		Electrical Conductivity (S/cm)	Response Time (s)	Recovery Time (s)	Ref.
	BET Surface Area (m^2^/g)	Pore Diameter (nm)				
PANI/NFC/PVA	17	17.1	—	47	58	[12]
PANI/CNF/PVA	34	21.2	—	41	50	[12]
PANI/CHT/PVA	—	—	3.9·10^−6^	—	—	[77]
PANI/NFC/PLA	—	—	—	—	—	[203]
PANI/CHN	—	—	2.2·10^−5^	120	—	[10]
PANI/CMC	—	—	7.6·10^−4^	10	90	[13,124]
PANI/MCC	—	—	1.25·10^−2^	40	60	[204]
PANI/NC	—	—	0.65	—	—	[116]
PANI/BC	—	—	0.12	—	—	[118]
PANI/EBC	—	—	1.1	—	—	[118]
PANI/EBC/PAM	—	—	1.4	—	—	[118]
PANI/CA	—	—	4.0·10^−3^	—	—	[123]

NFC—nanofibrillated cellulose, CNF—carbon nanofibers, PVA—polyvinyl alcohol, CHT—chitosan, PLA—polylactic acid, CHN—chitin, CMC—carboxymethylcellulose, MCC—microcrystalline cellulose, NC—nanocellulose, BC—bacterial cellulose, EBC—epoxy-modified bacterial cellulose, PAM—polyacrylamide, CA—cellulose acetate.

## Data Availability

Not applicable.

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
