# Peer review of "Polyaniline/Biopolymer Composite Systems for Humidity Sensor Applications: A Review"

_polymers, 2021, doi:10.3390/polym13162722_

Round 1

Reviewer 1 Report

The work entitled “Polyaniline/Biopolymer Composite Systems for Humidity Sensor Applications: A Review” by Anisimov et al. provides a detailed examination of the literature on polyaniline/biomaterial composites for humidity sensor applications. The authors focused on the last decade of research, trying to implement as much as possible examples of the last couple of years. The information provided focuses on the different perspectives of this research, offering a complete coverage of the subject. The data is well put together, organized, and follows a train of thought that is easily comprehended by the readers, which is very smart from the authors. The conclusions are very to the point and concise, with future perspectives on the subject being provided and leading the readers to the potential of these strategies. Overall, the manuscript has scientific merit (the authors did a great job) and is very up-to-date, and, thus, should be considered for publication after English revision is implemented. The only small lacuna to this work is the English writing that in some instances is a bit neglected and should be properly fixed (grammar should be more carefully cured and the use of more formal writing, in some instances, should be optimized). Further, there are some yellow highlights that should be eliminated.

Author Response

Author Response to Reviewer comments on MS ID:  polymers-1315406

Reviewer #1

The work entitled “Polyaniline/Biopolymer Composite Systems for Humidity Sensor Applications: A Review” by Anisimov et al. provides a detailed examination of the literature on polyaniline/biomaterial composites for humidity sensor applications. The authors focused on the last decade of research, trying to implement as much as possible examples of the last couple of years. The information provided focuses on the different perspectives of this research, offering a complete coverage of the subject. The data is well put together, organized, and follows a train of thought that is easily comprehended by the readers, which is very smart from the authors. The conclusions are very to the point and concise, with future perspectives on the subject being provided and leading the readers to the potential of these strategies. Overall, the manuscript has scientific merit (the authors did a great job) and is very up-to-date, and, thus, should be considered for publication after English revision is implemented. The only small lacuna to this work is the English writing that in some instances is a bit neglected and should be properly fixed (grammar should be more carefully cured and the use of more formal writing, in some instances, should be optimized). Further, there are some yellow highlights that should be eliminated.

Response: Corresponding edits have been carried out to address the reviewer comments in the revised manuscript. In particular, the authors have carried out comprehensive language editing throughout the manuscript to address grammar, syntax and clarity issues throughout to meet the high standards of this journal, Polymers. Related to the yellow (now changed to grey shading) highlights in Table 2, this was done deliberately to emphasize the most stable and most important form of PANI (generally, all humidity sensors are produced on its base).

We acknowledge Reviewer #1 for the valuable constructive and insightful comments on this manuscript.

Reviewer 2 Report

The authors have presented a comprehensive review that contains information on the materials based on polyaniline combined with some other polymers of natural origin as a background for humidity sensing.  The topic of investigation seems very interesting though the authors should better explain the necessity in such sensors and their advantages over alternative solutions described in the literature. In addition to the description of various materials classified in accordance with the nature of second (third) component, mechanisms of conductivity and hydration are considered in a scale obviously exhausting the demands of the topic of review announced. Briefly, this resulted in rather bulky and sometimes irregular data presentation and numerous duplications of the phrases. Some of the figures and tables  require careful reconsideration from the point of view of their necessity. The authors should also clearly explain what humidity sensors are demanded and what advantages  call for development of new materials. A high volume of the manuscript requires reduction of parts which seem not adequate to the topic of consideration. More details are presented below.

Abstract: Polyaniline acronym (PANI) should be introduced at first appearance of the term (line 13)

Abstract: First sentence sounds confusing: importance of biopolymers for humidity sensing is not obvious

Abstract: Line 18 – Please use PANI instead of “polymer-based materials”

Introduction, line 73: Please explain what’ PANI hybrid’ means

Figure 12 discussion: Please use PANI in the legend instead of polyaniline (this has no relation to the queries actually used); ‘chitaline’ is mentioned in three publication – less than the number of lines devoted to this term – please shorten or remove explanation.

Line 111 : “PANI has catalyzed…” – in this aspect, “promoted” sounds better

Table 1 – please move dimensions of variables to the first column, where possible. Please explain why the lines started with “measured” and “Price” are marked as bold

Lines 129-133: there are duplications in two sentences, please remove

Line 139 and below through the text – proton or hydrogen ion?

Lines 139 – 153: it is not clear why shuttle mechanism is presented only for hydroxide ions (Eq. 2) but not for hydrogen ions

Table 2 – please check whether the acronyms were already introduced (chitin as example) or not and leave description of the latter ones.

Lines 168-170 “The three classes of material types (polymer, ceramic and polymer/ceramic) depend on whether the sensing material is comprised of several components: organic polymers, inorganic ceramics or their mixtures.” – trivial statement, can be removed.

Figure 2 and table 3 – definition of ‘x’ value should be provided earlier – in the legend to Fig. 2. Please explain necessity of yellow color for emeraldine – there are only three lines in the table.

Lines 218-219: “In addition, water vapor impacts conductivity as well [67], which (?) can produce…” – please rephrase, sounds confusing

Lines 220-239 – Discussion of Table 2 should be moved to the Table 2. It looks rather strange to give first the description of Table 3 and then return to Table 2 again.

Line 228: any relative value varies from 0 to 100%, please explain or remove.

Lines 234-235 “nanoelectronic devices, including humidity sensors” – do humidity sensors belong to nanoelectronics? always?

Section 2 – there is no need to extend the review on the expense of the carriers different from PANI

Line 280 and below – there is no need to explain again resistive and capacitive mechanisms

Lines 3347-344. Please check the use of chitin an chitosan acronyms – they were already introduced above.

Figure 5 is trivial and should be removed

Figures 7, 19 – scale bar is invisible

Lines 485-486 “Oxidized units are characterized by the quinoid forms (=N–), whereas the reduced fraction consist of benzenoid units (–NH–), and the half-oxidized PANI is a mixture of both forms (=N– and –NH–)” – duplication of Fig. 2 discussion

Eq. (5) – add bonds to nitrogen atoms

Please give the reference to Fig. 9. Otherwise, add scale of CHT and PANI units to clarify the hopping possibilities.

Line 5143 “There is no known PANI/CHT humidity sensor to date.” - please give a reason to consider such a material in this review!

Table 5 does not contribute to better understanding of the information presented in the review

Table 6 – what is the sense to add H2S and CO2 sensors in the Table?

Line 731 “is depends” - please check

Line 772 “versus formation of H2O-H2O interactions” – formation of interactions?? please check and rephrase

Figure 27 – This is the Nyquist (not impedance) plot

Lines 996-997 “The intrinsic conductivity of PANI makes it possible for fabrication of smart electric devices such as humidity sensors.” – duplication, please remove

The description of factorial analysis (especially  Tables 8 and 9) id excessive.

Table 10 contains to many columns with few data (%, moisture adsorption) which should be removed. Check application of acronyms introduced elsewhere.

Author Response

Author Response to Reviewer comments on MS ID:  polymers-1315406

Reviewer #2

The authors have presented a comprehensive review that contains information on the materials based on polyaniline combined with some other polymers of natural origin as a background for humidity sensing.  The topic of investigation seems very interesting though the authors should better explain the necessity in such sensors and their advantages over alternative solutions described in the literature. In addition to the description of various materials classified in accordance with the nature of second (third) component, mechanisms of conductivity and hydration are considered in a scale obviously exhausting the demands of the topic of review announced. Briefly, this resulted in rather bulky and sometimes irregular data presentation and numerous duplications of the phrases. Some of the figures and tables  require careful reconsideration from the point of view of their necessity. The authors should also clearly explain what humidity sensors are demanded and what advantages  call for development of new materials. A high volume of the manuscript requires reduction of parts which seem not adequate to the topic of consideration. More details are presented below.

Abstract: Polyaniline acronym (PANI) should be introduced at first appearance of the term (line 13)

Response: Corrected.

Abstract: First sentence sounds confusing: importance of biopolymers for humidity sensing is not obvious

Response: The reviewer query was addressed in the revised manuscript.

Abstract: Line 18 – Please use PANI instead of “polymer-based materials”

Response: The reviewer query was addressed in the revised manuscript.

Introduction, line 73: Please explain what’ PANI hybrid’ means

Response: The term “hybrid” refers to composites that contain both synthetic polymers and biopolymer components, according to the revised manuscript.

Figure 12 discussion: Please use PANI in the legend instead of polyaniline (this has no relation to the queries actually used);

Response: The reviewer query was addressed in the revised manuscript.

 ‘chitaline’ is mentioned in three publication – less than the number of lines devoted to this term – please shorten or remove explanation.

Response: The reviewer is correct in terms of the wide adoption of the term “chitaline”. Notwithstanding the usage of the term, a more important aspect of “chitaline” pertains to the structure of PANI-CHT composites and the role of covalent versus ionic bonding in such materials. More importantly, we suggest that the discussion of PANI-g-CHT (or “chitaline”) composites should be retained due to its importance concerning the structure-function properties of such materials in PANI-chitosan humidity sensors, described in this contribution, as emphasized according to the variable structural forms of this system (cf. Fig. 6).

Line 111 : “PANI has catalyzed…” – in this aspect, “promoted” sounds better

Response: The reviewer query was addressed in the revised manuscript.

Table 1 – please move dimensions of variables to the first column, where possible. Please explain why the lines started with “measured” and “Price” are marked as bold

Response: The reviewer query was addressed in the revised version of Table 1.

Lines 129-133: there are duplications in two sentences, please remove

Response: The reviewer query concerning duplication of statements was addressed in the revised manuscript to avoid redundancy.

Line 139 and below through the text – proton or hydrogen ion?

Response: The reviewer query concerning the use of proton (hydrogen ion) was corrected in the revised manuscript.

Lines 139 – 153: it is not clear why shuttle mechanism is presented only for hydroxide ions (Eq. 2) but not for hydrogen ions

Response: We agree with the reviewer query concerning the mechanism for hydrogen ions. The proton hopping mechanism was added and the equations were renumbered accordingly.

Table 2 – please check whether the acronyms were already introduced (chitin as example) or not and leave description of the latter ones.

Response: The reviewer query concerning Table 2 was addressed in the revised manuscript.

Lines 168-170 “The three classes of material types (polymer, ceramic and polymer/ceramic) depend on whether the sensing material is comprised of several components: organic polymers, inorganic ceramics or their mixtures.” – trivial statement, can be removed.

Response: We agree with the reviewer suggestion and have made revisions in the updated manuscript.

Figure 2 and table 3 – definition of ‘x’ value should be provided earlier – in the legend to Fig. 2. Please explain necessity of yellow color for emeraldine – there are only three lines in the table.

Response: We agree with the reviewer and have made corresponding edits for Fig. 2 and Table 3.

Lines 218-219: “In addition, water vapor impacts conductivity as well [67], which (?) can produce…” – please rephrase, sounds confusing

Response: We agree with the reviewer and have made corresponding edits in the revised manuscript.

Lines 220-239 – Discussion of Table 2 should be moved to the Table 2. It looks rather strange to give first the description of Table 3 and then return to Table 2 again.

Response: We agree with the reviewer and have made corresponding edits for Table 2.

Line 228: any relative value varies from 0 to 100%, please explain or remove.

Response: We agree with the reviewer that some explanation is required. The statement of “0–100%” is important here since many devices are not quantitative across the full range of humidity values, as indicated in the explanation in the revised manuscript.

Lines 234-235 “nanoelectronic devices, including humidity sensors” – do humidity sensors belong to nanoelectronics? always?

Response: To address the reviewer query, the statement has been revised in the updated manuscript.

Section 2 – there is no need to extend the review on the expense of the carriers different from PANI

Response: The reviewer raises an important point. While the current review is focused on PANI-based materials, there is an important linkage to be made regarding the succession of materials between PANI and ceramic humidity sensors. Therefore, we contend that the inclusion of this material will serve the benefit of the wide readership of the journal, Polymers.

Line 280 and below – there is no need to explain again resistive and capacitive mechanisms

Response: The reviewer raises a valuable comment. Since the journal Polymers is not a specialist journal focused on electrochemistry, the authors suggest that a brief discussion of the resistive/capacitive mechanisms is warranted, particularly when the sensor material has variable conductivity and when greater precision is required in the case of capacitive sensors.

Lines 3347-344. Please check the use of chitin an chitosan acronyms – they were already introduced above.

Response: The manuscript was check for the use of such acronyms throughout and updated accordingly.

Figure 5 is trivial and should be removed

Response: The use of Fig. 5 serves the following intent: to highlight the facile synthesis of PANI-based composites and the utility of the use of biopolymer additives and the role of such green chemistry design strategies for humidity sensor materials. Therefore, we contend that Fig. 5 should be retained as emphasized in this review for such sensor materials.

Figures 7, 19 – scale bar is invisible

Response: The scale bar visibility has been addressed in the revised manuscript by adding larger scale bars in the revised version.

Lines 485-486 “Oxidized units are characterized by the quinoid forms (=N–), whereas the reduced fraction consist of benzenoid units (–NH–), and the half-oxidized PANI is a mixture of both forms (=N– and –NH–)” – duplication of Fig. 2 discussion

Response: The issued raised by the reviewer was addressed in the revised manuscript to minimize unnecessary duplication.

Eq. (5) – add bonds to nitrogen atoms

Response: The bonds were added in the revised eq. (5).

Please give the reference to Fig. 9. Otherwise, add scale of CHT and PANI units to clarify the hopping possibilities.

Response: The scaling was added to Fig. 9 to address the reviewer concern in the revised version.

Line 5143 “There is no known PANI/CHT humidity sensor to date.” - please give a reason to consider such a material in this review!

Response: The rationale for consideration of such PANI/CHT humidity sensors was added in the revised manuscript.

Table 5 does not contribute to better understanding of the information presented in the review

Response: The use of Table 5 was to draw a parallel between biomaterial-based sensors and carbon-based additives with PANI. A similar progression was drawn for ceramics vs biomaterials. In this regard, we contend that Table 5 is important in light of the future perspectives, conclusions, and related knowledge gaps in the field for this emerging class of organic humidity sensor materials.

Table 6 – what is the sense to add H2S and CO2 sensors in the Table?

Response: The intent for inclusion of such different types of carbon materials relates to the similar features of such sensors systems as it relates to other gases in parallel with humidity sensing. This is in line with the use of capacitive and resistive sensing modalities. 

Line 731 “is depends” - please check

Response: The corresponding edit was made in the revised manuscript.

Line 772 “versus formation of H2O-H2O interactions” – formation of interactions?? please check and rephrase

Response: The corresponding edits were made to address the reviewer query.

Figure 27 – This is the Nyquist (not impedance) plot

Response: The corresponding correction was made in the revised manuscript.

Lines 996-997 “The intrinsic conductivity of PANI makes it possible for fabrication of smart electric devices such as humidity sensors.” – duplication, please remove

Response: The corresponding duplicate statement was removed.

The description of factorial analysis (especially Tables 8 and 9) id excessive.

Response: We agree with the reviewer. The description regarding Tables 8 and 9 were revised, where Table 9 was completely removed and a citation was given to refer to the related data from the literature.

Table 10 contains to many columns with few data (%, moisture adsorption) which should be removed. Check application of acronyms introduced elsewhere.

Response: Table 10 was modified and unnecessary columns were removed.

In summary, the authors wish to acknowledge Reviewer #2 for the valuable constructive feedback and insightful comments on this manuscript. The revised manuscript was further edited for language, clarity and syntax throughout to meet the high standards of the journal Polymers.

Round 2

Reviewer 2 Report

I am fully satisfied with the authors' response and explanations. The manuscript can be recommended for publication.